**DOI: 10.1038/ncomms16019**　　**OPEN**

# Highly variable recurrence of tsunamis in the 7,400 years before the 2004 Indian Ocean tsunami

Charles M. Rubin[1,2], Benjamin P. Horton[1,2,3], Kerry Sieh[1,2], Jessica E. Pilarczyk[4], Patrick Daly[1], Nazli Ismail[5] & Andrew C. Parnell[6]

The devastating 2004 Indian Ocean tsunami caught millions of coastal residents and the scientific community off-guard. Subsequent research in the Indian Ocean basin has identified prehistoric tsunamis, but the timing and recurrence intervals of such events are uncertain. Here we present an extraordinary 7,400 year stratigraphic sequence of prehistoric tsunami deposits from a coastal cave in Aceh, Indonesia. This record demonstrates that at least 11 prehistoric tsunamis struck the Aceh coast between 7,400 and 2,900 years ago. The average time period between tsunamis is about 450 years with intervals ranging from a long, dormant period of over 2,000 years, to multiple tsunamis within the span of a century. Although there is evidence that the likelihood of another tsunamigenic earthquake in Aceh province is high, these variable recurrence intervals suggest that long dormant periods may follow Sunda megathrust ruptures as large as that of the 2004 Indian Ocean tsunami.

[1] Earth Observatory of Singapore, Nanyang Technological University, 639798 Singapore, Singapore. [2] Asian School of the Environment, Nanyang Technological University, 639798 Singapore, Singapore. [3] Department of Marine and Coastal Sciences, Rutgers University, New Brunswick, New Jersey 08901, USA. [4] Division of Marine Science, University of Southern Mississippi, Stennis Space Center, Hattiesburg, Mississippi 39529, USA. [5] Department of Physics/ Geophysics, Faculty of Mathematic and Natural Sciences, Syiah Kuala University, Banda Aceh 23111, Indonesia. [6] School of Mathematics and Statistics, Insight Centre for Data Analytics, University College Dublin, Belfield, Dublin 4 D04 V1W8, Ireland. Correspondence and requests for materials should be addressed to C.M.R. (email: cmrubin@ntu.edu.sg).

Projections of fatalities due to catastrophic earthquakes and tsunamis will likely exceed 2 million lives in the twenty-first century[1]. Advances in geodesy and seismology have contributed to our understanding of rupture patterns of large earthquakes, but the devastation caused by the 2011 Tohoku-oki and the 2004 Indian Ocean tsunamis make it clear that estimates of earthquake size and tsunami potential are woefully inadequate. The repeat times of such giant tsunamis can occur centuries to millennia apart[2–5] and are not fully captured in historical and instrumental records[2,3]. A more refined understanding of the long-term variations in timing and recurrence of giant tsunamis is essential for producing realistic vulnerability assessments for coastal communities.

The great Sumatra–Andaman earthquake triggered a tsunami that devastated south and southeast Asia[5,6]. At the time, there was no known historic precedent for the 1,500 km rupture of the Sunda megathrust[5], with slip exceeding over 20 m (refs 6,7). In the decade since the Indian Ocean tsunami, the search for prehistoric estimates of earthquake recurrence and tsunami potential remains elusive. Most reconstructions of past tsunami inundation are based on identifying anomalous beds of sand in low-energy environments, such as salt and freshwater marshes, coastal lakes or swales[8,9]. Prehistoric tsunamis have been identified using such geological records from northern Sumatra[10–12], Thailand[13–17], Andaman Islands[18], Sri Lanka[19], Eastern India[20] and the Maldives[21], but the timeline of their reconstructions is limited or fragmentary, hindered by preservation problems, reworking and a lack of accommodation space[22].

We identify coastal caves as a new depositional environment for reconstructing tsunami records and present a 5,000 year record of continuous tsunami deposits from a coastal cave in Sumatra, Indonesia (Fig. 1), which shows the irregular recurrence of 11 tsunamis between 7,400 and 2,900 years BP. The sedimentary record in the cave shows that ruptures of the Sunda megathrust vary between large (which generated the 2004 Indian Ocean tsunami) and smaller slip failures. The chronology of events suggests the recurrence of multiple smaller tsunamis within relatively short time periods, interrupted by long periods of strain accumulation followed by giant tsunamis. The data demonstrates that the 2004 tsunami was just the latest in a sequence of devastating tsunamis stretching back to at least the early Holocene and suggests a high likelihood for future tsunamis in the Indian Ocean. The sediments preserved in the costal cave provide a unique opportunity to refine our understanding of the behaviour of the Sunda megathrust, as well as study in detail the sedimentology and hydrological characteristics of tsunami deposits.

## Results

**Geologic setting.** The coastal cave site is located along the northwestern coast of Aceh Province near the village of Lhong, 35 km south of Banda Aceh (Fig. 2). This segment of the Sunda megathrust (Fig. 1) slipped as much as 20 m during the 2004 rupture[6,7] and produced nearly 1 m of subsidence. The 2004 tsunami inundated the cave and removed vegetation off the very steep limestone cliff to a height of ∼24 m above mean tidal level (MTL) which was over 10 m above the top of the cave entrance (Fig. 2). The cave entrance is 100 m back from the swash zone with a rock sill at its entrance that sits 1 m above mean tidal level (Fig. 2). The cave extends nearly 120 m into the cliff. We excavated six trenches at the rear of the cave (Fig. 2) and found sedimentary sequences up to 2 m thick above a limestone basement.

**2004 tsunami deposit.** The 2004 tsunami deposited a sand bed in all trenches, which was 20–43 cm thick. The 2004 tsunami sand

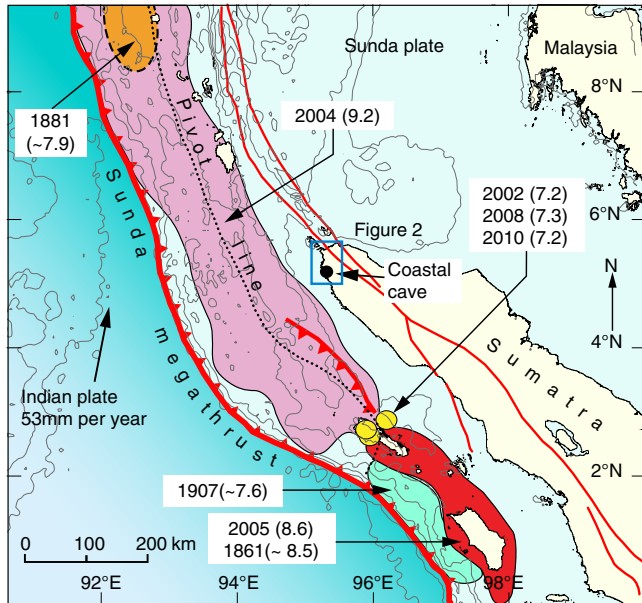

**Figure 1 | Tectonic setting and ruptures of major earthquakes along the Sunda megathrust.** The pink patch is the estimated rupture area of the 2004 Indian Ocean earthquake[6]. The red patch is the estimated rupture area of the 2005 Nias–Simeulue earthquake[60]. Orange and green patches show the area of the 1881 and 1907 earthquakes. Yellow circles show the location of the 2002, 2008 and 2010 eathquakes. Solid lines depict primary faults generalized from Singh et al.[61]. Pivot line shows location of uplift and subsidence of the seafloor during the 2004 earthquake[5]. Relative plate motion is from Prawirodirdjo and Bock[62].

bed is laterally continuous, well-sorted, composed of fine to very fine grained sand. In the trench nearest the cave entrance (Trench 6), the 2004 tsunami sand bed has three pulses of coarse material followed by subsequent fining upwards sequences (Fig. 3; Supplementary Fig. 1; Supplementary Tables 1 and 2). Basal rip-up clasts, lenticular laminations and fragments of weathered cave chalk are common in the 2004 sand bed in all trenches. The 2004 sand bed contains abundant, pristine foraminifera, mostly of benthic subtidal origin[23], but with a notable planktonic offshore presence. Organic debris, transported into the cave by the tsunami and guano from the insect-feeding bats (*Microchiroptera*) that inhabit the cave, littered the surface of the 2004 tsunami deposit. The basal contact of the 2004 deposit is an erosional unconformity.

**Prehistoric tsunami deposits.** Beneath the 2004 tsunami deposit, we found an additional 11 sand beds (A–K) that we interpret as tsunami deposits (Fig. 4). The 11 sand beds consist of well-sorted, normally graded, very fine sand to silt with a sharp basal contact. There is no evidence of unconformities in the stratigraphic sequences from the trench-wall exposures (Supplementary Figs 1 and 2). Sand beds G–J have thin deposits (2–7 cm), whereas sand bed F has the thickest deposit (23 cm). Some of the sand beds have a rip-up clast-rich lower portion and a lenticular-laminated upper portion. The rip-up clasts are very similar to the deposits that underlie them. Large detrital weathered fragments of cave chalk are preserved in the sand beds. Foraminifera are abundant, in particular in sand beds I–K (Fig. 5; Supplementary Tables 3 and 4). The provenance of the foraminifera ranges from intertidal to subtidal to offshore[23]. A large percentage of the foraminiferal assemblage in each sand bed is pristine (Fig. 5).

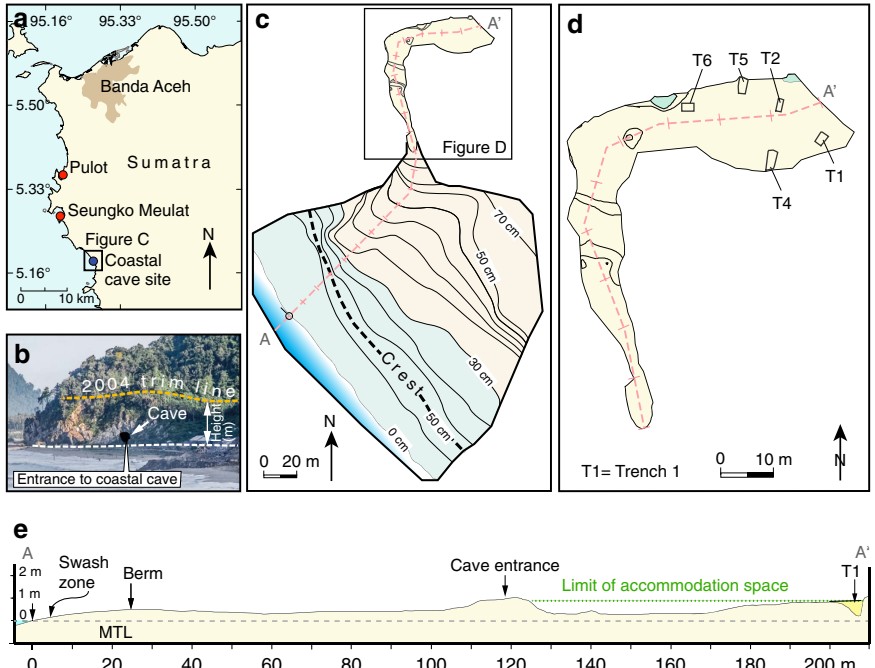

**Figure 2 | Map showing site locations and topography of the cave site.** (**a**) Location of coastal cave site and buried soils from the northwestern coast of Aceh Province[12]. (**b**) Photograph of the coastal cave entrance and the 2004 trim line. The trim line is about 10 m above the entrance to the cave. (**c**) Topographic map of the coastal cave site. (**d**) Map showing excavations in the coastal cave. (**e**) Topographic profile from the swash zone to the coastal cave.

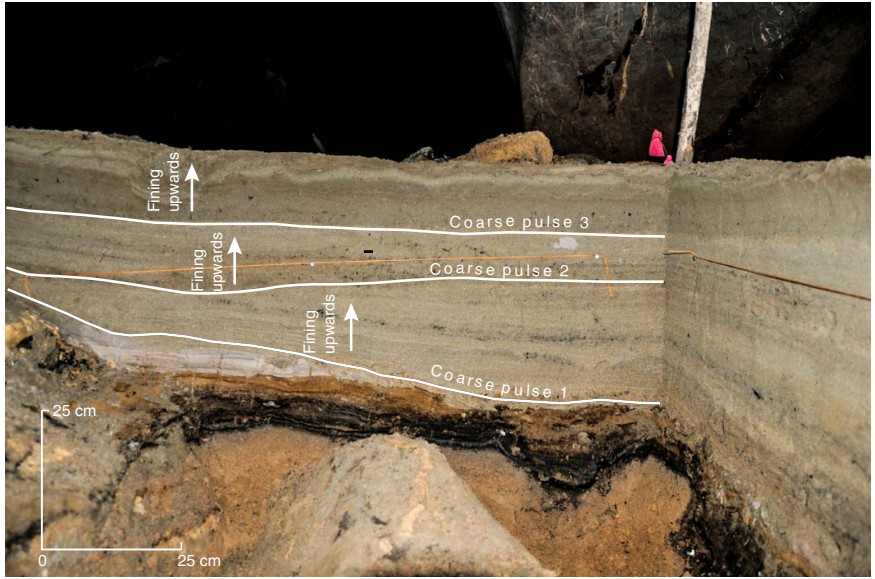

**Figure 3 | 2004 Indian Ocean tsunami deposit.** Photograph showing three coarse pulses and fining upwards sequences exposed in Trench 6. Grain size samples for the 2004 deposit collected from Trench 6 are shown in Supplementary Table 3. See Fig. 2 for location of Trench 6.

**Intercalated beds**. Organic-rich beds are found between the 11 sand beds (A–K), reflecting slow sediment accumulation during intervals between tsunamis. The organic beds are commonly finely laminated and range in thickness from <1 mm to 9 cm (Fig. 4; Supplementary Figs 1 and 2). The intercalated beds consist of sands, reworked by periodic drips of water through the cave ceiling during periods of high precipitation and insect burrowing. The organics were likely produced by the same processes that produce organic debris on the surface of the 2004 tsunami and have been broken down by post-depositional

processes. Foraminifera are absent or in low abundances with a fragmented and abraded assemblage of intertidal to subtidal to offshore species (Fig. 5), further suggesting the intercalated beds are reworked from the tsunami sand beds A–K. In many intercalated beds, we found small, pristine and fragile chalk florets.

Four mud beds appear between sand beds B–H with thicknesses up to 25 cm (Figs 4 and 5; Supplementary Figs 1 and 2). The upper contact of the mud beds is sharp and locally eroded, consistent with the presence of rip-up clasts within the overlying

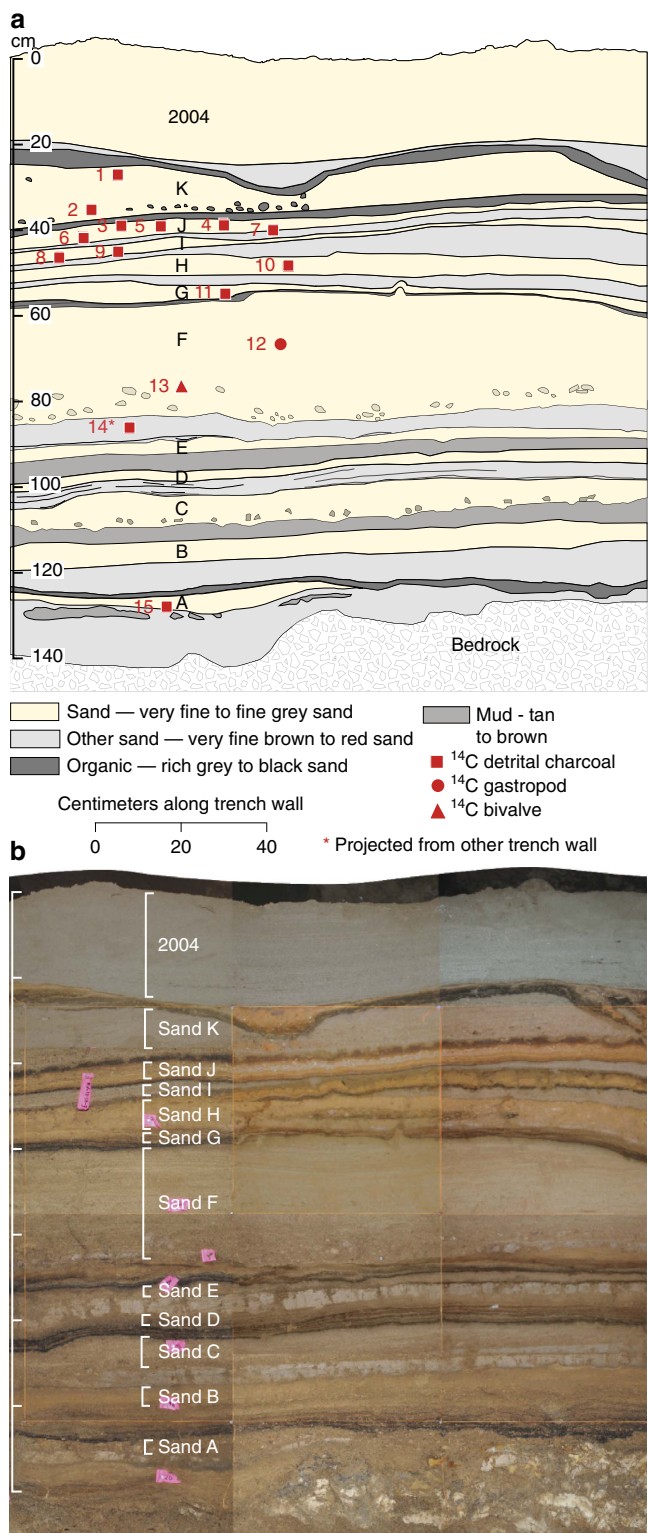

**Figure 4 | Coastal cave stratigraphic units and the tsunami sand beds.**
(**a**) Stratigraphic units showing the 11 tsunami sand beds (A–K) and the 2004 tsunami deposit. Red numbers refer to the radiocarbon samples (Table 1). (**b**) Photomosaic of Trench 1 showing location of the 11 sand beds, A–K. See Fig. 2 for location of Trench 1.

tsunami sand beds. In two mud beds foraminifera are present in low abundances, dominated by , intertidal assemblages (Fig. 5; Supplementary Tables 3 and 4). However, foraminifera are absent in the other two mud beds, suggesting deposition by freshwater

ponding in the cave. Pristine cave chalk is found in some mud beds, further supporting deposition in low-energy conditions.

**Chronological constraints**. We obtained accelerator mass spectrometry (AMS) radiocarbon ages on pieces of detrital charcoal and whole molluscs from within, below and above the sand beds (Fig. 6; Table 1). We interpret the two bracketing dates as maximum and minimum ages for the timing of sand bed deposition. Fragments of charcoal from an organic-rich bed at the base of the sedimentary sequence yield a maximum age of 7,672–7,588 years BP for sand bed A. Charcoal from a mud bed (5,583–5,331 years BP) is the maximum age for sand F. A pristine mollusc shell within the sand bed F provide an age of 5,258–4,552 years BP. Charcoal yield ages of 3,362–3,246 years BP and 3,363–3,245 years BP for sand beds G and H, respectively. Multiple charcoal dates from sand beds I, J and K provide age ranges of 3,366–3,221 years BP, 3,464–3,068 years BP and 2,975–2,772 years BP, respectively.

## Discussion
Coastal caves have not previously yielded prehistoric records of tsunamis. Indeed, the cave's sheltered location and absence of human activity suggest that these sand beds represent the best-preserved and most complete tsunami history for the Indian Ocean between 7,400 and 2,900 years BP. The cave's interior protects the tsunami deposits from erosion. The rock sill near the cave entrance (Fig. 2) mitigated the erosional impact of tsunamis that are found at elevations beneath the sill. However, deposits above the rock sill are vulnerable to scouring from subsequent events. The cave's location also disfavours sand bed deposition or re-working by intense storms[12,14]. Exposure to tropical cyclones is limited due to the lack of Coriolis force near the equator[24,25]. In addition, the track of any tropical cyclones that originate in Indian Ocean will move towards India, Bangladesh or Myanmar without producing a storm surge in Sumatra[14,26]. Although tropical cyclones do strike eastern Thailand, they dissipate after crossing the Malay Peninsula and Sumatra before moving offshore along Sumatra's west coast (for example, tropical storm Vamei in 2001 (refs 27,28)).

The stratigraphic and microfossil data of the 11 prehistoric sand beds (A–K) resemble the 2004 tsunami as well as tsunami deposits described elsewhere. Rip-up clasts at the base of sand beds and sharp basal contacts suggest erosion occurred at the beginning of the tsunami inundation as the surge entered the cave. Normally graded sand beds indicate settling from suspension following tsunami inundation in the cave[9,29]. The normal grading suggests that each bed resulted from a single (rather than multiple) instance of the cave filling with water and draining. The foraminifera assemblage of the sand beds were dominated by intertidal to subtidal to offshore species. Marine foraminifera often dominate tsunami deposits because of the landward transport and deposition of scoured marine sediment[23,30]. The taphonomic (or surface) condition of individual foraminifera distinguishes the tsunami sand beds and the intercalated beds (Fig. 5). The foraminifera of the tsunami sand beds is predominantly pristine suggesting the foraminifera were entrained from a protected subtidal substrate[31,32].

We have also identified cave chalk weathering as a new indicator of tsunami inundation. Large fragments of weathered cave chalk are preserved in the sand beds. These fragments most likely fell from the cave ceiling and were weathered due to abrasion by tsunami transport. In contrast, we found pristine and fragile chalk florets in the organic and mud beds.

Holocene relative sea-level reconstructions from the Indo-Pacific region are characterized by a mid-Holocene sea-level high stand of

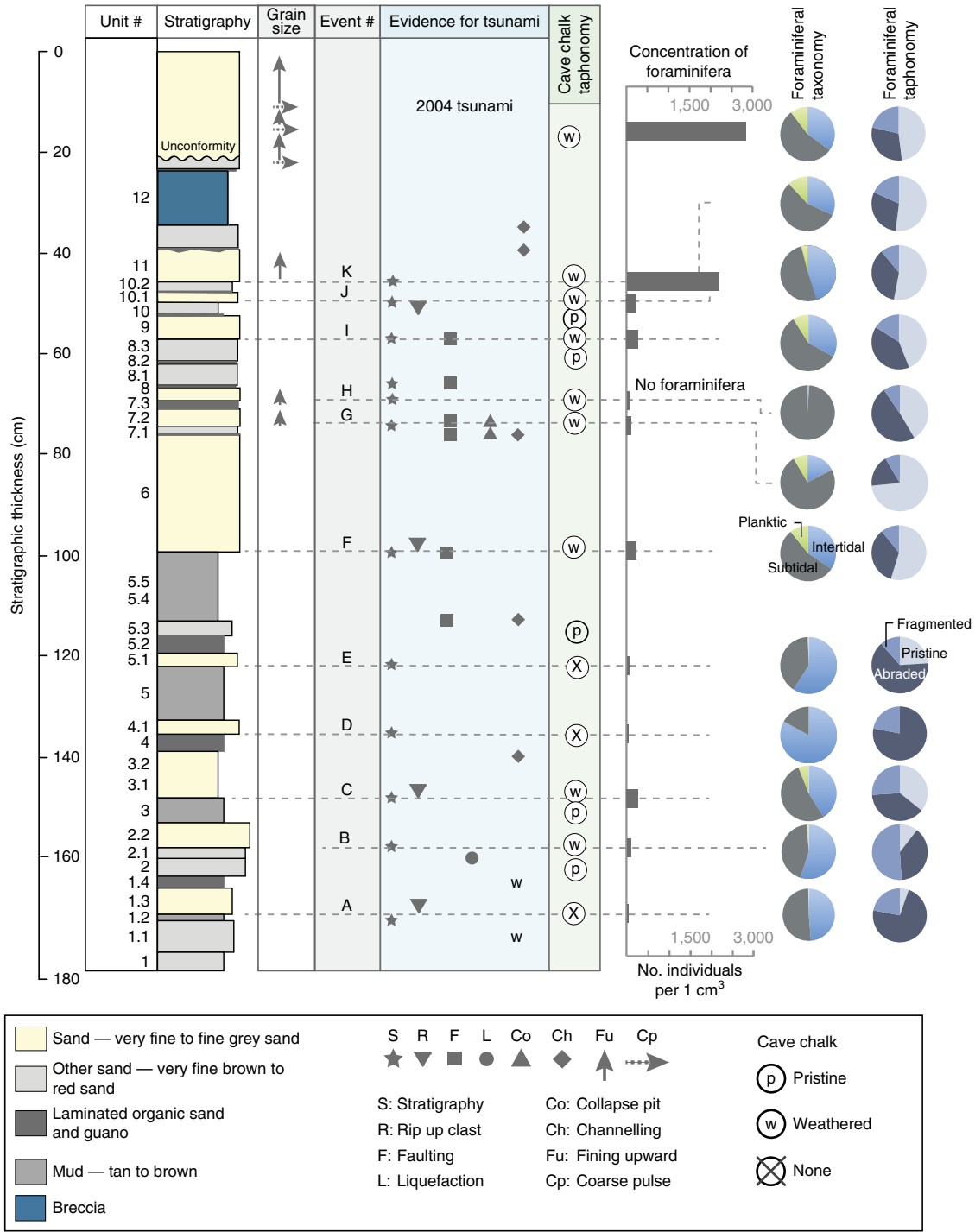

**Figure 5 | Stratigraphic and microfossil evidence for tsunamis.** Vertical arrows show fining upwards sand and horizontal arrows indicate coarse sand pulse. Average thickness of units taken from Trench 1 vertical exposures, trench walls A, B, C and D. Grain size from Supplementary Table 1; foraminifera data from Supplement Table 2.

a few decimetres to several metres[33,34], but the presence or absence of such a highstand may be controlled by local tectonic processes[35]. A record of buried soils from the northwestern coast of Aceh Province suggest that relative sea-level rose during the early and mid-Holocene from −5 m at ∼7,900 years BP to −1.6 m at ∼5,700 years BP[12]. Relative sea-level was below present until at least 3,800 years BP. In the late Holocene, relative sea-level stabilized within 0.4 m of modern sea-level[12,22,33]. This gradual long-term relative sea-level rise without a mid-Holocene highstand

created a time-window for tsunami deposits and intercalated beds to aggrade without a significant interruption in sedimentation[36] (Fig. 7).

The cave probably contained stratigraphic evidence of recent historic tsunamis from 2,900 years BP to the 2004 Indian Ocean tsunami that have been identified elsewhere in the region[10,14], but these were most likely removed by subsequent tsunamis inundating the cave as indicated by the erosional unconformity beneath the 2004 deposit (Figs 4 and 7; Supplementary Fig. 1).

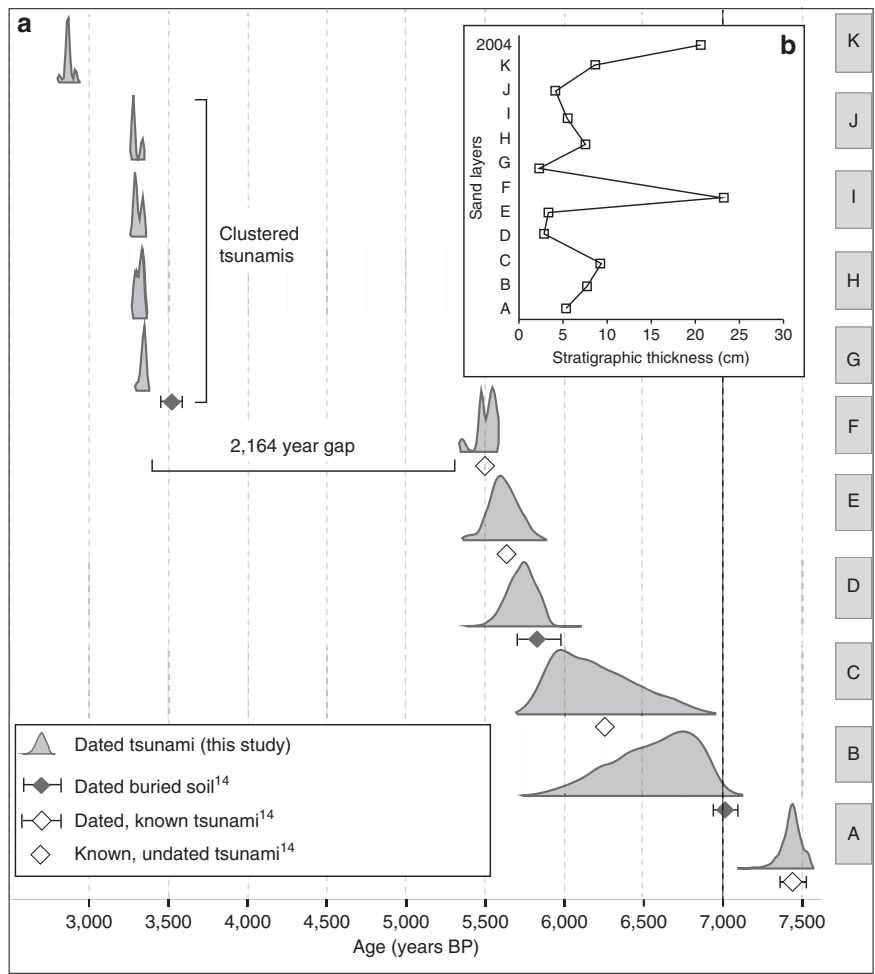

**Figure 6 | Bayesian model showing timing of tsunamis. (a)** We use a custom Bayesian model for sand beds A–K that simultaneously calibrates all radiocarbon dates, incorporates the law of superposition and the constraints of limiting dates, which lie between or beyond the range of the directly dated tsunamis (Methods section). The model is fitted using a Markov chain Monte Carlo approach[55,63]. Radiocarbon ages calibrated with Calib rev. 6.0.0 (ref. 53), age ranges appear with 95.4% highest density region (HDR) ($\sim$2 s.d.), where years 'before present' (BP) is years before A.D. 1950 (Table 1). **(b)** Sediment thickness of the sand beds and the 2004 Indian Ocean tsunami deposit. The stratigraphic thickness is the average thickness across 29 vertical sections in Trench 1 Faces A and C (Supplementary Table 5).

The missing stratigraphic record coincides with the continued aggradation of the nearby coastal plain[12]. Stratigraphical records of late Holocene tsunamis are generally restricted to environments with sufficient accommodation space, such as intervening coastal swales between ridges[10,14], estuaries and ponds where overwash deposits are protected from erosion by rapid growth of vegetation or deposition of sediment[36].

Independent evidence for tsunami inundation in the cave comes from stratigraphy of nearby coastal lowlands[12,37]. Three coseismic subsidence events and seven tsunamis between $\sim$7,500 and 3,800 years BP are documented in the stratigraphy of the west coast of northern Sumatra[12]. During this time interval, the cave sequence preserves an identical number of tsunamis (that is, sand beds A–G; Fig. 6 and Table 1). Offshore of Sumatra, Patton *et al.*[38] identified 11 deep-sea turbidites along the Andaman–Aceh slip patch between $\sim$6,500 and 2,700 years BP. Although the number of events is the same, the timing of the events is different. The deep-sea turbidite record does not capture the tightly clustered tsunamis (sand beds G–J) and the large gap in time between tsunamis F and G. The discrepancies suggest that ruptures along the Sunda megathrust do not always trigger both tsunami deposits and turbidites.

Since the 2004 tsunami, considerable evidence for prehistoric tsunamis has been obtained from sites around the Indian Ocean[13–16,18,20,21,39–43]. However, studies with time spans comparable to the cave are restricted to Sri Lanka[19] and the Maldives[21]. In southern Sri Lanka, Jackson *et al.*[19], identified seven tsunami sand beds between $\sim$6,700 and 2,400 year BP. Klostermann *et al.*[21] identified three tsunami sand beds between $\sim$5,600 and $\sim$2,900 years BP in the Maldives. However, these far-field records do not capture the tightly clustered tsunamis (sand beds G–J) and have events that span the large gap in time between tsunami sand beds F and G. The far-field seismic sources for these tsunamis are uncertain. For example, slip along the megathrust near the Andaman–Nicobar Islands are potential seismic sources for tsunamis in Sri Lanka and the Maldives[44]. In addition, the faults along the southern coast of Pakistan are potential seismic sources for tsunamis in the Maldives[44]. The immediate proximity of the cave to the Sunda megathrust provides a more reliable indicator of tsunamis generated by ruptures of the megathrust than far-field records.

The chronology from accelerator mass spectrometry (AMS) radiocarbon ages from the sand beds of the coastal cave and the stratigraphy of the nearby coastal lowlands[12,37], combined within a

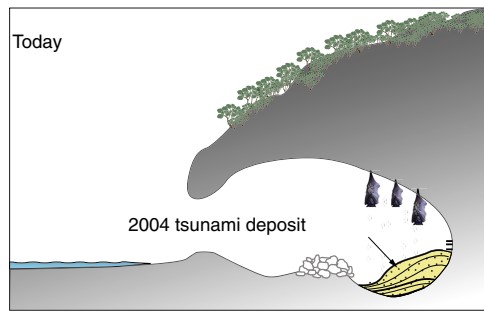

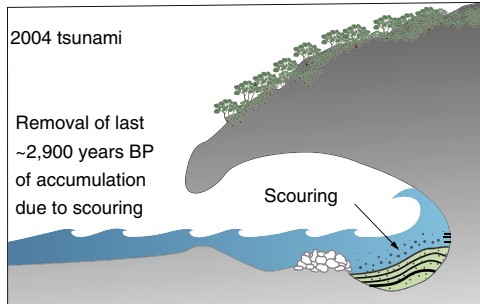

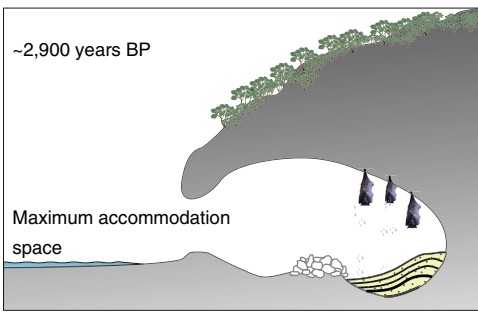

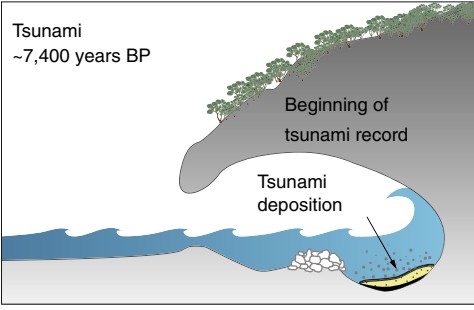

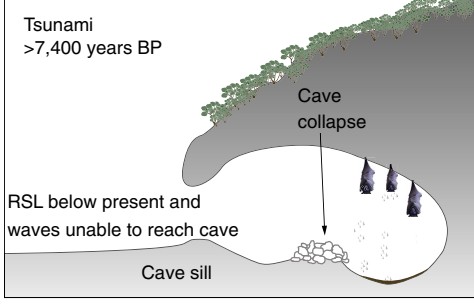

**Figure 7 | Model of tsunamis deposits in cave interior.** Our model shows the development of accommodation space and the accumulation of stacked tsunami beds A–K and the 2004 Indian Ocean tsunami in the coast cave.

Bayesian framework (Fig. 6; Methods section), provides the chronology of tsunamis between 7,400 and 2,900 years BP (Table 1). The chronology suggests an average recurrence interval

of 456 years between 7,400 and 2,900 years BP with a large uncertainty (95% C.I. 1–2,210) (Fig. 6; Supplementary Table 7). A similar average recurrence interval (600–900 years) was estimated from the nearby coastal lowlands of northwestern Aceh Province between 7,400 and 3,800 years BP[12]. The tsunami record from Sri Lanka[19] suggested an average recurrence interval of ∼360 years between 6,600 and 4,200 year BP.

The tsunami record from the cave, however, indicates a dramatic variation in recurrence interval. Between 7,400 and 5,500 years BP, the recurrence interval for tsunamis A to F was 681 years (95% C.I. 11–2,222) (Fig. 6). But after 5,500 years BP, the coastal cave has an age gap of 2,164 years (95% C.I. 1,997–2,247) between tsunamis F and G. Four tightly clustered tsunamis (G–J) occurred between 3,400 and 3,300 years BP with an average recurrence interval of 16 years (95% C.I. 0–55). The most recent tsunami (K) recorded in the coastal cave occurred at 2,900 years BP, with a recurrence interval of 426 years (95% C.I. 357–505). Although the time span of the northern Simeulue coral microatoll record is limited to the last millennium, it shows a similar large variation of recurrence intervals from 56 years to ∼550 years[39].

There is a correlation between the thickness of tsunami sand beds and recurrence intervals in the cave (Fig. 6). The thinner sand beds (G–J) have the smallest recurrence intervals and were preceded by the largest age gap between tsunamis (Supplementary Fig. 4). The thickest sand bed (F) preceded the large age gap, with a thickness similar to the 2004 tsunami sand bed. Although variations of offshore sediment availability or lateral shoreline changes might play a role in the thickness of tsunami beds[45], we suggest that the thickness of the sand beds may reflect the size of slip along the megathrust. It is possible that sand bed F was deposited by a giant tsunami produced by a large slip that was followed by a very long dormant, interseimic period with substantial strain accumulation. Subsequently, partial, smaller slip failures occurred in rapid succession between 3,400 and 3,300 years BP, producing sand beds G–J. The very long dormant period suggests that the Sunda megathrust is capable of accumulating large slip deficits between earthquakes. Such a high slip rupture would produce a substantially larger earthquake than the 2004 event.

The dramatic variation in tsunami recurrence intervals suggests a continuum of recurrence behaviour from large slip ruptures (2004 tsunami and sand bed F), earthquake super cycles or doublet earthquake[2,4,11,39] to smaller slip failures (for example, sand beds G–J) similar to the October 2010 Mentawai tsunamigenic earthquake (Fig. 1). Variations in recurrence may result from temporal changes in coupling or locking depth, or very long-term slow non-tsunamigenic slip events[46,47]. If thickness of the tsunami deposit (Fig. 6) reflects slip and the size of the slip patch, the thickness of the 2004 tsunami deposit implies a long dormant period until the next large slip event. Slow non-tsunamigenic slip events might predominate during such long periods of quiescence and precede clustering smaller slip failures along the megathrust. The remarkable variability of recurrence suggests that regional hazard mitigation plans should be based upon the high likelihood of future destructive tsunami demonstrated by the cave record and other paleotsunami sites, rather than estimates of recurrence intervals.

## Methods

**Field methods and data collection.** The evidence for prehistoric tsunamis is derived principally from stratigraphic relations found in six trench-wall exposures and other smaller pits in the cave. The stratigraphy is visually striking, because of a strong contrast in colour between alternating beds of sand, mud and laminated organic sand (Supplementary Figs 1 and 2).

We identified and described stratigraphic units in Trench 1 and the trench walls were mapped from high-resolution photos of the vertical surfaces. The alternating beds of sand, clay and laminated organic sand allow stratigraphic correlation of individual beds within the trench-wall exposure (Supplementary Figs 1 and 2). We divided the stratigraphy into 12 units, from Unit 1, which is the oldest and deepest,

**Table 1 | Radiocarbon analyses of charcoal and shells.**

| Sand layer | No. | Unit | Material dated | Laboratory code | $^{14}$C age (1σ error) | $\delta^{13}$C | Calibrated (95.4% HDR) | Age of event (years BP) | Notes |
|---|---|---|---|---|---|---|---|---|---|
| K | 1 | 11 | Charcoal | MnL-12-4D-1 | 2,822 ± 20 | −25.47 | 2,862–2,975 | 2,815–2,916 | Age of K |
|  | 2 | 11 | Wood | MnL-12-1D-5 | 2,725 ± 20 | −26.22 | 2,772–2,859 |  | Age of K |
| J | 3 | 10.1 | Charcoal | MnL-12-1A-3 | 2,965 ± 20 | −26.71 | 3,068–3,236 | 3,270–3,341 | Age of J |
|  | 4 | 10.1 | Charcoal | MnL-12-4A-1 | 3,065 ± 20 | −25.38 | 3,219–3,356 |  | Age of J |
|  | 5 | 10.1 | Charcoal | MnL-12-1C-2ii | 3,093 ± 21 | −25.85 | 3,260–3,370 |  | Age of J |
|  | 6 | 10.1 | Charcoal | MnL-12-1C-2i | 3,210 ± 21 | −25.89 | 3,383–3,464 |  | Age of J |
|  | 7 | 10 | Charcoal | MnL-12-1C-1 | 3,217 ± 21 | −28.06 | 3,269–3,396 |  | Age of J |
| I | 8 | 9 | Charcoal | MnL-12-4C-2 | 3,085 ± 21 | −26.24 | 3,252–3,366 | 3,278–3,346 | Age of I |
|  | 9 | 9 | Charcoal | MnL-12-1C-3 | 3,069 ± 20 | −26.95 | 3,221–3,358 |  | Age of I |
| H | 10 | 8.2 | Charcoal | MnL-12-4A-2 | 3,078 ± 21 | −28.45 | 3,245–3,363 | 3,287–3,353 | Age of H |
| G | 11 | 8 | Charcoal | MnL-12-4C-4 | 3,077 ± 20 | −26.17 | 3,246–3,362 | 3,304–3,363 | Age of G |
|  |  | UBS* | Wood | SM 11 13A 182 | 3,540 ± 30 | −29.74 | 3,717–3,902 |  | Maximum age of G |
| F | 12 | 6 | Gastropod | MnL-12-4-5 | 4,666 ± 43 | −9.75 | 4,552–5,258 | 5,357–5,575 | Age of F |
|  | 13 | 6 | Bivalve | MnL-12-4-12 | 4,638 ± 43 | −9.38 | 4,513–5,231 |  | Age of F |
|  | 14 | 5.4 | Charcoal | MnL-12-1D-4 | 4,742 ± 23 | −26.78 | 5,331–5,583 |  | Maximum age of F and minimum age of E |
| E |  |  |  |  |  |  |  | 5,480–5,770 |  |
| D |  | MBS* | Wood | PU 07 04 265 | 5,090 ± 40 | −28.30 | 5,743–5,917 | 5,578–5,866 | Maximum age of D and minimum age of C |
| C |  |  |  |  |  |  |  | 5,857–6,680 |  |
| B |  | LBS* | Wood | PU 07 03 426 | 6,060 ± 40 | −25.80 | 6,791–7,142 | 6,083–6,915 | Maximum age of B and minimum age of A |
| A | 15 | 1.1 | Charcoal | MnL-12-1C-5 | 6,788 ± 26 |  | 7,588–7,672 | 7,324–7,529 | Maximum age of A |
|  |  | LBS* | Wood | SM 11 13 490 | 6,560 ± 35 |  | 7,424–7,558 |  | Maximum age of A |

LBS, lower buried soil; MBS, middle buried soil; UBS, upper buried soil.
Samples are listed in stratigraphic order. Calibrated age ranges (year BP) at 95.4% HDR, using Calib rev. 6.0.0 (ref. 53) and a ΔR value of 15 ± 119 for the marine reservoir effect[54]. Analytical uncertainties are 1σ and reflect the total uncertainty in the measurement. The age of event combines the AMS radiocarbon ages from the sand beds of the coastal cave and the stratigraphy of the nearby coastal lowlands[12,37], within a Bayesian framework. The 5–95% confidence limits are shown.
*UBS; MBS; LBS; dates from stratigraphic sequences in Kelsey et al.[12]

resting on the limestone bedrock of the cave, through Unit 12, which is the youngest unit that underlies the 2004 Indian Ocean tsunami sand bed (Fig. 5). Column samples were taken across each stratigraphic interface, providing samples for grain size, microfossil and chronology.

Topographic data were based on a total station survey in 2011 and 2012 (Fig. 2). Excavation of six trenches within the cave interior was conducted in 2011 and 2012. We made measurements of sedimentology and stratigraphy in the field.

**Tsunami sand beds.** We use several lines of stratigraphic evidence that point to a tsunami origin for sand beds A–K that includes stratigraphic relations, lithology, degree of sorting, internal structures, such as fining upwards beds, rip-up clasts of mud, sharpness of stratigraphic contacts, uniformity in bed thickness and lateral continuity of beds[8,48–50]. In addition, we use secondary features such as liquefaction, bedding-plane faults (décollements) and normal faults that are perhaps triggered by seismic shaking[8].

We analysed foraminiferal assemblages (Fig. 5; Supplementary Tables 3 and 4) from most units to confirm marine inundation and indicate the provenance of the overwash sands[32]. In addition, we used foraminiferal test taphonomy[51] to reveal depositional and post-depositional environmental conditions (Supplementary Fig. 3; Supplementary Tables 3 and 4). The foraminiferal analyses help constrain sediment provenance and help identify tsunami sands in a variety of environmental settings.

We used the taphonomy of cave chalk as a new indicator of the environment of deposition. The cave chalk found in the sedimentary units most likely fell from the cave ceiling due to weathering. The chalk occurs in two forms: (1) small, fragile, pristine florets; and (2) large detrital rounded fragments that do not have a delicate floret structure (Supplementary Fig. 3). Since the florets do not show evidence of water transport (for example, rounding), we suggest that the delicate florets are limited to sand beds not deposited by tsunamis. In contrast, the rounded chalk fragments are only found in sand beds that were produced by a tsunami. We believe the rounding of the fragments is due to abrasion by the tsunami waves.

Grain size and thickness data allows comparison of the 2004 tsunami sediments with the prehistoric sand beds (Figs 5 and 6; Supplementary Table 1). Further, the general fining upwards trends in many tsunami beds suggests deposition by tsunami waves (Fig. 5; Supplementary Table 1). Following an initial high water flow, a decreased flow velocity often causes sand beds to deposit in graded, fining upwards sequences[9,52]. For example, along the Aceh coastline, the 2004 tsunami sand bed records distinct fining upwards sequences of sand deposition[52], similar to the 2004 tsunami sand bed preserved in the coastal cave. Samples for foraminifera, cave chalk and grain size analyses are from Trenches 1 and 4 (Fig. 5; Supplementary Tables 1, 2, 3 and 4).

**Radiocarbon dating.** We collected detrital charcoal fragments from seven stratigraphic units (Units 1, 5, 8, 9, 10 and 11), and two intact gastropods from Unit 6. The radiocarbon ages constrain ages of tsunamis; calibrated radiocarbon ages helped constrain the timing of tsunamis. Detrital charcoal and gastropods were collected from units for radiocarbon dating and were analysed by GNS, Rafter Radiocarbon Laboratory, New Zealand (Table 1).

We calibrated radiocarbon ages with Calib rev. 6.0.0 (ref. 53). The calibrated age ranges appear with 95.4% HDR (~2 standard deviations), where years 'before present' (BP) is years before A.D. 1950 (Table 1). Also, we corrected the ages of the gastropod shells for the marine reservoir effect using a ΔR value of 15 ± 119 (ref. 54) to account for the fact that Indian Ocean waters show substantial $^{14}$C depletion due to upwelling.

To further constrain chronology, we analysed the organic guano beds derived from insect-feeding bats that occupy the coastal cave. We sampled six organic-rich beds along thin stratigraphic horizons (3–6 mm). Although the radiocarbon analyses indicate the dark sand beds are broadly mid-late Holocene in age, three radiocarbon dates are not consistent with stratigraphic position. We suggest that the discrepancies of the bulk guano dates are due to: (1) bulk samples containing an unknown mixture of organic material of variable age, and representing an average age of the sample; and (2) groundwater percolating along cracks in the limestone cave, and introducing exogenous, old carbon into the organic beds.

**Bayesian age-depth model.** We apply a Bayesian age-depth model to 19 radiocarbon dates from the coastal cave and the nearby coastal lowlands[12]. The radiocarbon dates either directly date a tsunami or provide maximum or minimum age limits for a tsunamis (Fig. 6; Table 1) (Supplementary Tables 6 and 7). Our Bayesian modelling approach provides control over the model fitting process and flexibility in the modelling assumptions. The code is available at https://github.com/andrewcparnell/tsunamis.

We use the following notation to build our model. $\theta_i$ is the calendar age of tsunami $i$, where $i$ runs from 1 to 11. These are the parameters we are most interested in estimating. Together, we write these values as $\theta$. $x_{ij}$ is the direct radiocarbon date $j$ of tsunami $i$, where $j = 1,\ldots,n_i$ with $n_i$ the number of direct dates for tsunami $i$. These values have associated fixed 1-sigma errors $\sigma_{ij}$. Note, that for some tsunamis there are no direct dates, in which case $n_i = 0$. Thus, while we have 13 direct dates in total, five tsunamis are without direct dates. Together, we write

these values as $x$. $\theta_i^*$ are the calendar ages of limiting dates lying between tsunamis $i$ and $i+1$. They are nuisance parameters. Together, we write these values as $\theta^*$. $x_{ij}^*$ is a limiting radiocarbon date $j$ lying between tsunamis $i$ and $i+1$. These values have associated fixed 1-sigma errors $\sigma_{ij}^*$. Here $j = 1, \ldots, n_i^*$, where $n_i^*$ represents the number of limiting dates for tsunami $i$. As above, some of these are 0. Together, we write these values as $x^*$. $\gamma_i$ are the calendar age shifts of the limiting dates which provide a maximum age of tsunami $i$. They are nuisance parameters which we write together as $\gamma$. $x_{ij}^{**}$ is a limiting radiocarbon date $j$ for tsunami $i$, providing evidence of a maximum age. These values have associated fixed 1-sigma errors $\sigma_{ij}^{**}$. Here $j = 1, \ldots, n_i^{**}$, where $n_i^{**}$ represents the number of limiting dates for tsunami $i$. Again, some of the $n_i^{**}$ values are 0, as we have only three such dates. Together we write these values as $x^{**}$. Note that the calibrated value of a radiocarbon age $x_{ij}^{**}$ is $\theta_i + \gamma_i$, that is, the calendar age of the tsunami plus a shift indicating how much older the radiocarbon date is beyond that of the tsunami itself. $r(\theta)$ is the IntCal13 calibration curve which has the probability distribution $r(\theta) \sim N(\mu(\theta), \tau^2(\theta))$. We assume that both $\mu()$ and $\tau^2()$ are known functions.

Our overall goal is to find the posterior distribution:

$$\pi(\theta, \theta^*, \gamma \,|\, x, x^*, x^{**}, r) \propto \prod_{i=1}^{n} \prod_{j=1}^{n_i} \pi(x_{ij}|\theta_i) \times \prod_{i=1}^{n} \prod_{j=1}^{n_i^*} \pi(x_{ij}^*|\theta_i^*) \times \prod_{i=1}^{n} \prod_{j=1}^{n_i^{**}} \pi(x_{ij}^{**}|\theta_i, \gamma_i) \times \pi(\theta) \times \pi(\gamma),$$

(1)

where

$$x_{ij}\big|\theta_i \sim N\left(\mu(\theta_i), \sigma_{ij}^2 + \tau^2(\theta_i)\right)$$

(2)

$$x_{ij}^*\big|\theta_i^* \sim N\left(\mu(\theta_i^*), \left(\sigma_{ij}^*\right)^2 + \tau^2(\theta_i^*)\right)$$

(3)

$$x_{ij}^{**}\big|\theta_i, \gamma_i \sim N\left(\mu(\theta_i + \gamma_i), \left(\sigma_{ij}^{**}\right)^2 + \tau^2(\theta_i + \gamma_i)\right)$$

(4)

are the likelihood terms. The prior distribution $\pi(\theta)$ is set to enforce the ordering of the dates:

$$\pi(\theta)\big(\theta_1 > \theta_1^* > \theta_2 > \theta_2^* > \ldots > \theta_{11}\big),$$

(5)

where $I$ is an indicator function. The prior distribution on the excesses $\gamma_i$ is the only informative prior in the model. We use:

$$\gamma_i \sim Ga(1, 0.005),$$

(6)

which corresponds to a gamma distribution with a mean of 200 and a s.d. of 200. While this distribution is diffuse, it is informative for the maximum ages. The distribution corresponds to a 95% probability that the maximum ages are no more than 600 years older than the tsunami they are aiming to represent.

Our model is fitted with Markov chain Monte Carlo using Metropolis-Hastings steps for all parameters since ordering the tsunami dates was complicated by their prior distribution[55]. The model is sensitive to starting values of the parameters due to the ordering constraint, so we simulate suitable values by calibrating dates individually, and sampling from these distributions with an extra restriction on the ordering. We run the model with multiple different starting values, and check convergence using trace plots and the Geweke convergence diagnostic[55]. The final model run created 1 million iterations, removing 100,000 for a burn-in period and then keeping only every 450th iteration.

**2004 tsunami sand bed.** The 2004 tsunami sand bed ranges from about 20 to 43 cm in thickness in trench-wall exposures (Fig. 3; Supplementary Figs 1 and 2). It is a light grey, normally graded fine to very fine sand (mean = 2.7Φ; % sand = 91.9%), with abundant laminations, some of which can be traced more than a metre.

In Trench 6, the 2004 tsunami sand bed records three distinct beds (Fig. 3), delineated by three pulses of coarse material followed by subsequent fining upwards sequences (Fig. 3; Supplementary Table 1). We interpret each combined coarse pulse and fining upwards sequence as individual tsunami waves. The first coarse pulse at the base of the 2004 sand (41–42 cm) is marked by an influx of fine to medium sand (mean = 2.1Φ; % sand = 94.1%), which fines upwards until 34 cm (mean = 2.7Φ; % sand = 90.6%). The second coarse pulse occurs from 31–33 cm (mean = 2.3Φ; % sand = 95.6%) and fines slightly up to 24 cm (mean = 2.7Φ; % sand = 92.5%). The final coarse pulse is located between 22 and 24 cm (mean = 2.5Φ; % sand = 96.3%), and fines upwards to a very fine sand (mean = 3.7Φ; % sand = 74.0%) at the top of the sequence (0–1 cm).

Rip-up clasts, consisting of organic-rich granules, wood and shells are common, especially in the lower part of the 2004 deposit. Abundant (2,500–3,246 individuals per 1 cm³) foraminifera that are predominantly pristine (41–52%) and sourced from subtidal (52–58%), intertidal (33–38%) and offshore (for example, planktic) (8–14%) environments are present, as are weathered fragments of cave chalk (Fig. 5, Supplementary Fig. 3; Supplementary Table 4). The 2004 sand has the highest diversity foraminiferal assemblage, with *Pararotalia* sp., *Amphistegina* sp. and *Calcarina* sp. dominating. A sharp and erosional contact marks the boundary between the 2004 tsunami sand and the underlying Unit 11. The erosional removal of pre-2004 sediment is variably preserved in trenches and on the cave walls as alternating remnants of sand and organic-rich sand.

**Unit 1 sand bed A.** The lowest stratigraphic unit (Unit 1) above the limestone cave floor is an irregular, laminated, dark, organic layer. Overlying the Unit 1 organic layer is fine sand (Unit 1.1; mean = 3.0Φ; % sand = 77.4%) that does not contain foraminifera or cave chalk. Unit 1.2 is discontinuous marine-influenced clay (grain size data not available) that pinches out toward the corner of Trench 1. The clay bed contains relatively low numbers of foraminifera (20 individuals per 1 cm³). The species assemblage is dominated by intertidal (71%) and subtidal (29%) species, with a paucity of planktic and deeper-dwelling benthic foraminifera. The taphonomic assemblage is dominated by abraded (52%) and fragmented (48%) individuals, with no pristine individuals present (Fig. 5; Supplementary Table 1). Unit 1.2 is devoid of cave chalk.

Unit 1.2 is overlain by a 5.4 cm thick fine sand (Unit 1.3; mean = 2.9Φ; % sand = 74.9%) with sparse laminations and abundant rip-up clasts derived from the underlying clay. The lower stratigraphic contact between Unit 1.3 and the underlying clay is sharp (~2 mm), along an erosional and irregular surface. On the basis of the abundance of rip-up clasts, we interpret Unit 1.3 as the oldest tsunami bed, labelled Sand Bed A. Unit 1.3 sand contains a low number of foraminifera (21–62 individuals per 1 cm³) consisting predominantly of subtidal (38–58%) and intertidal (41–62%) species. The assemblage is dominated by the subtidal species *Epinoides* sp., *Cibicides lobatulus* and *Pararotalia* spp., with a near absence of planktics. Individual foraminifera were both abraded (68–77%) and fragmented (14–30%), with only 2–9% in pristine condition (Fig. 5; Supplementary Table 3). Overlying Unit 1.3 (sand bed A) is a laminated, dark, organic layer (Unit 1.4; mean = 2.3Φ; % sand = 85.0%); the contact between the units is gradational over a few centimetres.

Angular fragments of detrital charcoal from Unit 1.1 yielded a calibrated age range of 7,650–7,510 cal. years BP (Table 1), which we interpret as the maximum age of Sand Bed A or Unit 1.3.

**Unit 2 sand bed B.** Units 2.0 and 2.1 consist of a dark, red to brown sand with laminations of varying thickness. The grain size is fine sand (Units 2 and 2.1: mean = 2.6Φ; % sand = 85.1%). The units contain abundant delicate (pristine) cave chalk florets (Fig. 5; Supplementary Fig. 3) that suggest a non-tsunami source for the sand. These units have moderate abundances of foraminifera (68–94 individuals per 1 cm³), and are dominated by intertidal species (55–66%), which are mostly abraded (49–58%) and fragmented (31–34%) (Fig. 5; Supplementary Tables 3 and 4), and probably reworked from Sand Bed A (Unit 1.3).

Overlying Unit 2.1 is a ~6 cm thick fining upwards, fine to very fine sand (Unit 2.2; mean = 3.5Φ; % sand = 75.2%; Supplementary Fig. 1; Supplementary Table 1). Along the base of Unit 2.2, pebble-sized clay rip-up clasts are abundant. Locally, Unit 2.2 grades into a thin, discontinuous dark sand. Unit 2.2 contains weathered cave chalk and moderate abundances of foraminifera (76–89 individuals per 1 cm³) from subtidal (41–47%) and intertidal (51–59%) environments (Fig. 5; Supplementary Tables 3 and 4). Unlike the underlying Unit 2.1 foraminifera-bearing sediments, species within Unit 2.2 (Fig. 5) include the deeper-dwelling *Lagena* sp., and *Operculina ammonoides*. The rip-up clasts, normal grading and the presence of subtidal foraminifera and weathered cave chalk in Unit 2.2 imply deposition by a tsunami (sand Bed B). However, the foraminiferal assemblage of sand bed B, unlike most other candidate tsunami sand beds in the cave sequence, is mostly fragmented (47–55%), and contains only minor abundances of pristine individuals (9–12%). The contact between Unit 2.2 and the overlying Unit 3 clay is sharp (~2 mm), but irregular.

**Unit 3 sand bed C.** The base of Unit 3 consists of a 3.6 cm thick marine-influenced clayey mud (mean = 4.2Φ; % clay = 11.9; % silt = 34.7). Although in low abundance (32–55 individuals per 1 cm³), the presence of foraminifera from intertidal (46–51%) and subtidal (49–54%) environments, many of which were pristine (43–45%), implies deposition in a quiet intertidal environment. Cave chalk at the bottom of Unit 3 is absent, but present as pristine florets at the top. Overlying Unit 3 is a 9.2 cm thick, slightly normal graded, fine sand (Unit 3.1: mean = 2.0 Φ; % sand = 87.5%) which grades into a massive to faintly laminated fine sand over a few centimetres (Unit 3.2; mean = 2.5Φ; % sand = 85.9%) (Fig. 5; Supplementary Table 1). The base of Unit 3.1 contains abundant pebble-sized angular rip-up clasts from underlying mud (mean = 4.2Φ; % sand = 53.3%). Scouring of Unit 3 clayey mud before deposition of Unit 3.1 is clear in several places in Trench 1. The sand of Units 3.1 and 3.2 contains weathered cave chalk and abundant (146–412 individuals per 1 cm³) foraminifera that are predominantly pristine (32–41%) and from subtidal (46–61%), intertidal (37–45%) and planktic (2–9%) environments (Fig. 5; Supplementary Table 3). Dominant species include *Elphidium craticulatum*, *Cibicides lobatulus* and *Epinoides repandus*. The abundance of rip-up clasts, graded bedding, subtidal foraminifera and weathered cave chalk suggests a tsunami origin for Units 3.1 and 3.2, labelled Sand Bed C. Overlying Unit 3.2 is a laminated fine organic-rich sand (Unit 4: mean = 2.3Φ; % sand = 90.2%) that is devoid of foraminifera and cave chalk, and has a sharp (~2 mm) but irregular contact.

**Unit 4 sand bed D.** The base of Unit 4 consists of a 4-cm thick, laminated, organic-rich sand. The laminations are clear but discontinuous, and vary from black to dark red-brown. Thickness also varies markedly, from <1–12 cm. The

sand lacks foraminifera and cave chalk. Overlying Unit 4 is a 3-cm thick, fine sand (Unit 4.1; mean $= 2.5\Phi$; % sand $= 86.8\%$; Fig. 5, Supplementary Table 1). The stratigraphic contact between Unit 4.1 fine sand and the underlying Unit 4 is sharp (a few mm), along an irregular surface. Unit 4.1 contains a low number of foraminifera (19 individuals per 1 cm$^3$). Many of the tests are abraded (78%) and fragmented (22%), and no pristine individuals were found. Where present, foraminifera were of intertidal (83%; *Ammonia convexa, Ammonia parkinsoniana*) or subtidal (17%; *Pararotalia* sp.) origin (Fig. 5; Supplementary Table 4). No cave chalk was observed in Unit 4.1. On the basis of the presence of fine sand, with a basal erosional contact and subtidal foraminifera in Unit 4.1, we infer a speculative tsunami, labelled Sand Bed D (Supplementary Fig. 1). Overlying Unit 4.1 is a clayey mud (Unit 5) with a sharp ($\sim 2$ mm), but irregular contact.

**Unit 5 sand bed E.** Unit 5 consists of an extensive non-marine clayey mud. The clayey mud is 11 cm thick, but varies in thickness from 4 to 15 cm. The massive clay contains pristine cave chalk, but no foraminifera and was likely deposited by ponding water in the cave resulting from periods of increased precipitation. The upper contact of the clay is smooth and displays only minor, local evidence of erosion. Overlying the clayey mud is a massive 3-cm thick, very fine sand (Unit 5.1; mean $= 3.4\Phi$; % sand $= 72.9\%$), that has a low foraminiferal assemblage (35–42 individuals per 1 cm$^3$; Fig. 5, Supplementary Tables 1 and 3). The foraminifera are highly abraded (63–66%), and predominantly from an intertidal (*Ammonia parkinsoniana* and *Elphidium advenum*) environment (56–62%). Unit 5.1 is devoid of cave chalk. On the basis of the presence of massive sand and marine foraminifera, we speculate that Unit 5.1 (labelled Sand Bed E) represents deposition as a result of a tsunami. The absence of rip-up clasts from the underlying clay suggests that the flow of the water in the cave was too weak to erode the underlying clay. In addition, it is hard to explain how the massive sand could have been deposited from other processes so far back into the cave. Overlying Unit 5.1 is a 3-cm thick, laminated, very fine sand that varies from black, guano-rich lamina to reddish brown, inorganic lamina (Unit 5.2: mean $= 4.0\Phi$; % sand $= 58.3\%$). Along other trench wall exposures, a thin, massive, fine sand (Unit 5.3: no grain size data available) overlies Unit 5.1. The fine sand lacks rip-up clasts, and contains fragments of pristine cave chalk and a very low number (2 individuals per 1 cm$^3$) of abraded (100%) foraminifera, representing reworking of older units. The contact between Unit 5.1 (sand bed E) and the overlying units is irregular. Overlying the thin organic-rich fine sand (Unit 5.2) are the thickest non-marine clayey mud beds exposed in the cave sedimentary sequence (Units 5.4 and 5.5; no grain size data available) (Supplementary Figs 1 and 2). Their combined thickness ranges up to about 25 cm. The clayey mud is differentiated into two separate units based upon their colour: the upper part of the clayey mud (that is, Unit 5.3) is darker (more organic) than the lower part of the clay (Unit 5.4). Both units are devoid of foraminifera and were likely deposited by the same processes that formed Unit 5. The units lack cave chalk.

Fragments of detrital charcoal from the overlying Unit 5.4 clayey mud yielded a calibrated age range of 5,583–5,331 years BP, which we interpret as the minimum age of sand bed E.

Further evidence from secondary folding of Units 5.1 and 5.2 and the truncation of these units along the top of Unit 5.3 clayey mud suggests local ground shaking (Supplementary Fig. 2c). On the basis of the intact stratigraphic continuity of Units 5.3 and 5.4, ground shaking must have occurred before the deposition of Unit 5.3.

**Unit 6 sand bed F.** Overlying the clayey mud units is a 23-cm thick sequence of two sand beds (Unit 6) with rip-up clasts and subtidal and planktic foraminifera. Together, Unit 6 sand beds are the thickest in Trenches 1 and 4 (Fig. 6b; Supplementary Figs 1 and 2; Supplementary Table 5). Thicknesses range up to 30 cm in Trench 1 to about 20 cm in Trench 4. The lower sand of Unit 6 is a massive, fine sand (mean $= 2.9\Phi$; % sand $= 78.9\%$) with abundant pebble- and cobble-sized rip-up clasts from underlying Units 5.4 and 5.5 (Supplementary Figs 1 and 2; Supplementary Tables 1 and 2). The massive sand grades into a laminated fine sand (mean $= 2.7\Phi$; % sand $= 87.3\%$). The contact between Unit 6 and the underlying clayey mud (Unit 5.5) is sharp and erosional, over a few centimetres. Unit 6 contains subtidal (43–68%), intertidal (18–53%) and planktic (4–14%) foraminifera (198–296 individuals per 1 cm$^3$) (Fig. 5; Supplementary Tables 3 and 4), with *Elphidium advenum, Calcarina* spp. and *Operculina ammonoides* dominating the assemblage. While the Unit 6 lower sand contains weathered fragments of cave chalk, the upper sand is devoid of them. The abundance of rip-up clasts, graded bedding, subtidal foraminifera and weathered cave chalk suggests a tsunami origin for Unit 6, labelled sand bed F. Similarly to other sand units that represent tsunami deposits, including the 2004 tsunami deposit, the majority of foraminifera are generally pristine (41–69%). Overlying Unit 6 (Supplementary Figs 1 and 2; sand bed F) is a <2.0-cm thick, black, organic sand that grades into a heterogeneous, laminated, grey sand (Unit 7). The contact between Unit 6 and the overlying Unit 7 is sharp (a few millimetres) and irregular.

Two pristine gastropod shells from Unit 6 yielded a calibrated ages range of 5,231–4,515 years BP and 5,258–4,552 years BP, which we interpret as the age of Sand Bed F.

**Unit 7 sand bed G.** Underlying Unit 7.1 or Sand Bed G is a 1.5-cm thick, black, organic, fine sand (Unit 7: mean $= 2.6\Phi$; % sand $= 89.4\%$) with discontinuous laminations. Its thickness varies from a few millimetres to about 5 cm. The contact between Unit 7 and the overlying Unit 7.1 is sharp ($\sim 2$ mm) along an erosional surface. Unit 7.1 consists of irregularly laminated fine sand (mean $= 3.0\Phi$; % sand $= 79.8\%$) that ranges from 1 to 4 cm in thickness. Unit 7.1 fines upwards from fine sand at the base (mean $= 2.6\Phi$; % sand $= 85.1\%$) to very fine sand at the top (mean $= 3.6\Phi$; % sand $= 72.8\%$; Supplementary Fig. 2; Supplementary Tables 1 and 2). Unit 7.1 contains weathered cave chalk and foraminifera (93–105 individuals per 1 cm$^3$) that are dominantly pristine (65–82%) and subtidal (71–77%; Supplementary Fig. 2; Supplementary Table 4). *Pararotalia* sp. and *Cibicides* spp. dominate the foraminiferal assemblage. In places, the fine sand laminations are defined by heavy minerals. The normal grading, the presence of subtidal foraminifera and weathered cave chalk imply deposition by a tsunami (Fig. 5 sand bed G). Sand bed G (Unit 7.1) is overlain by a thin clayey mud (Unit 7.2: mean $= 4.5\Phi$; % clay $= 14.7$; % silt $= 34.6$) that is laterally discontinuous. Unit 7.2 is devoid of both foraminifera and cave chalk fragments. The contact is gradational over a few millimetres.

**Unit 8 sand bed H.** Overlying the clayey mud (Unit 7.2) is a $\sim 4.5$-cm thick, massive sand that upwardly fines from fine sand at the base (Unit 8; mean $= 2.3\Phi$; % sand $= 82.3\%$) to very fine sand at the top (mean $= 3.8\Phi$; % sand $= 71.7\%$; Supplementary Fig. 2; Supplementary Tables 1 and 2). Weathered fragments of cave chalk that are up to several centimetre in diameter are present in Unit 8 sand. Unit 8 contains moderate abundances (55–62 individuals per 1 cm$^3$) of foraminifera that are almost exclusively from subtidal environments (98–100%), 39–44% of which are pristine (Fig. 5; Supplementary Tables 3 and 4). Dominant species include the deeper-dwelling *Lagena* sp., *Operculina ammonoides* and *Pararotalia stellata*.

The abundance of graded bedding, subtidal foraminifera and weathered cave chalk suggests a tsunami origin for Unit 8, labelled sand bed H (Supplementary Figs 1 and 2). Overlying Unit 8 or Sand bed H is a thin, clayey mud bed (Unit 8.1: mean $= 4.6\Phi$; % sand $= 53.5\%$) that is only exposed in Trench 4. The contact between Unit 8.1 and the underlying sand bed H (Unit 8.1) is irregular, but sharp over a few millimetres.

Unit 8.2 consists of massive to laminated, very fine sand (mean $= 3.2\Phi$; % sand $= 82.9\%$). Unit 8.2 is devoid of foraminifera and cave chalk. Unit 8.3 consists of thin, black and grey to dark brown, organic, sandy-mud (mean $= 5.7\Phi$; % sand $= 19.8\%$), about 2 cm thick. The dark organic sandy-mud contains pristine cave chalk fragments and a low number (20 individuals per 1 cm$^3$) of foraminifera that are generally abraded (96%), and sourced from intertidal (55%) and subtidal (45%) environments. The contact between Unit 8.2 and the overlying Unit 9 sand is sharp and irregular.

Further evidence from a secondary high-angle, normal fault that disrupts Units 1 through 7 (Supplementary Fig. 1e), suggest that faulting occurred before deposition of Units 8 or sand bed H.

Fragments of detrital charcoal from Unit 11 yielded a calibrated age range of 3,362–3,246 years BP, which we interpret as the age of sand bed H.

**Unit 9 sand bed I.** Unit 9, consists of a light tan, massive, fine sand (mean $= 2.3\Phi$; % sand $= 86.3\%$), up to 3 cm thick, with weathered detrital fragments of cave chalk, particularly in the upper half of the unit (Supplementary Fig. 2; Supplementary Table 4). Angular fragments of detrital charcoal are also common. Faint laminations are present, in particular at the base of the unit. Unit 9 does, however, locally fill small scours cut into the underlying Unit 8.3 sandy-mud. This indicates minor erosion prior to deposition. Foraminifera are abundant (256 individuals per 1 cm$^3$), and are predominantly subtidal (58%) species (*Pararotalia stellata, Epinoides repandus* and *Cibicides refulgens*); $\sim 44\%$ of the foraminiferal assemblage is pristine (Fig. 5; Supplementary Tables 3 and 4). The presence of fine sand, subtidal foraminifera and weathered cave chalk imply deposition by a tsunami, labelled sand bed I (Fig. 5 band bed I). Unit 9 sand (band bed I) is overlain by a black, organic bed only a few millimetre thick (Unit 9.1: no grain size data), with a sharp contact. A low-abundance assemblage (4 individuals per 1 cm$^3$) consisting of abraded (82%), intertidal (54%), and subtidal (46%) foraminifera and fragments of pristine cave chalk is present in this unit, and were derived from underlying band bed I (Unit 9). Fragments of detrital charcoal from Unit 9 yielded two calibrated age ranges of 3,358–3,221 years BP and 3,366–3,252 years BP, which we interpret as the age of sand Bed I. In addition, fragments of detrital charcoal from the top of Unit 9 sand yielded a calibrated age range of 3,363–3,245 years BP, which we interpret as the maximum age of Sand Bed I.

Further evidence from truncation of Units 8.1, 8.2 and 8.3 suggests a bedding-parallel fault along the top of Unit 8 (Supplementary Fig. 1c). Secondary folding of these units above this bedding-parallel fault implies that displacement occurred after deposition of Unit 8.3 but before deposition of Unit 9 (sand bed I).

**Unit 10 sand bed J.** Unit 10 is a thin, very fine sand with red-brown laminations (mean $= 3.2\Phi$; % sand $= 75.6\%$). Unit 10 contains a low abundance (nine individuals per 1 cm$^3$) and weathered (76% fragmented and 24% abraded)

foraminiferal assemblage (86% intertidal species), with pristine cave chalk fragments. This unit represents accumulation of bat guano and remobilized sand from the underlying Unit 9 sand. Overlying the red, very fine sand of Unit 10, along a sharp contact, is a thin (2 cm thick), massive fine sand (Unit 10.1; mean = 2.8Φ; % sand = 85.9%; Supplementary Figs 1 and 2; Supplementary Tables 1 and 2), with abundant pebble-sized fragments of weathered cave chalk. Unit 10.1 sand contains abundant (190 individuals per 1 cm³) foraminifera that are predominantly pristine (53%) and from subtidal (51%) and intertidal (45%) environments, with only a small percentage of planktic species present (4%; Supplementary Fig. 2; Supplementary Tables 3 and 4). Dominant species include *Ammonia parkinsoniana*, *Elphidium craticulatum* and *Heterolepa* sp. The contact with the underlying Unit 10 is sharp. On the basis of the presence of very fine well-sorted sand and subtidal foraminifera, we infer that Unit 10.1 (Fig. 5; labelled sand bed J) represents deposition as a result of a tsunami. Overlying Unit 10.1 (sand bed J) is a black to dark brown, massive, organic, fine sand (Unit 10.2; mean = 2.5Φ; % sand = 87.8%). Unit 11 has been eroded away from most of the other trench exposures. Unit 10.2 is thin, black to dark brown, massive, organic, fine sand, about 1.5 cm thick, that represents bat guano accumulation and sand remobilization derived from Unit 10.1.

Fragments of detrital charcoal from Unit 10.1 yielded four calibrated ages: 3,464–3,383 years BP, 3,370–3,260 years BP, 3,356–3,219 years BP, 3,236–3,068 years BP. These calibrated ages represent the age of sand bed J. In addition, fragments of detrital charcoal from Unit 10 yielded a calibrated age range of 3,396–3,269 years BP, which we interpret as the maximum age of sand bed J.

**Unit 11 sand bed K.** Unit 11 is an 8.4-cm thick sequence that fines upwards from a medium sand (Unit 11; mean = 1.9Φ; % sand = 95.5%) at the base to a fine sand (mean = 2.4Φ; % sand = 94.3%) with numerous laminations at the top (Supplementary Figs 1 and 2; Supplementary Tables 1 and 2). The unit contains abundant rip-up clasts that are derived from the underlying Unit 10.2. The uppermost laminated, very fine sand is nearly absent in other excavations due to erosion, but it is preserved in most of the Trench 1 exposures (Supplementary Fig. 1). Weathered fragments of cave chalk are abundant and range in size up to 3 cm. Unit 11 also abounds in foraminifera (1,976–2,485 individuals per cm³), and 49–55% of the foraminifera are pristine. Subtidal (48–61%) species are dominant (*Pararotalia* sp., *Asterorotalia* sp. and *Epinoides repandus*) with planktic foraminifera present, but in smaller amounts (9–14%; Fig. 5, Supplementary Tables 3 and 4). On the basis of the normal grading, abundant rip-up clasts and subtidal foraminifera, we suggest that this sand layer represents a tsunami deposit (Fig. 4; labelled sand bed K). Sand bed K (Unit 11) is separated from the overlying black to grey, thin, laminated, fine sand (Unit 12: no grain size data available) by an erosional unconformity. Unit 12 is a massive, fine sandy pebble breccia with an irregular black, organic sand cap. Clasts range up to about 10 cm in length and are commonly angular.

Fragments of detrital charcoal from Unit 11 yielded two calibrated age ranges of 2,859–2,772 years BP and 2,975–2,862 years BP. We interpret these as the age of sand bed K.

**Grain size analysis.** We analysed grain size at 1 cm resolution with a Malvern MS 3000 laser particle size analyzer (measuring grain sizes up to 1,800 μm). Before analysis, we removed organics and carbonate with 30% hydrogen peroxide and 10% hydrochloric acid, respectively. We subsequently let the samples disaggregate in a sodium hexametaphosphate solution for 24 h.

We calculated grain size values with the Wentworth-Phi scale[56], using the average of three runs. Grain size descriptions for each sampled interval follow those defined by Blott and Pye[57], and include a mean (average grain size), mode (dominant grain size), s.d. (degree of sorting) and percentage of clay, silt and sand. We show the depths of coarse pulses and fining upwards sequences in Fig. 5.

**Foraminiferal and cave chalk analysis.** We examined 42 samples from Units 1–12 for foraminiferal taxa (Supplementary Table 8) and taphonomy (surface condition of foraminiferal tests). Foraminiferal taxonomy constrains provenance through ecology, and taphonomy can determine residence time and transport history. For foraminifera and cave chalk analysis, we subsampled 5 cm³ samples from each layer, wet-sieved them at >63 μm, dried at 25 °C and examined them under a binocular microscope. Sieved samples were dry split to obtain counts of ~300 foraminifera per sample[58]. For each sample, the total number of foraminifera present in 5 cm³ (total concentration) was calculated, as was the percentage of each species present (abundance). Foraminiferal taxonomy followed Loeblich and Tappan[59]. Foraminifera were further categorized according to the taphonomic (surface) condition of the tests as defined by Pilarczyk and Reinhardt[51], where pristine individuals are those that are taphonomically unaltered; abraded individuals are those that are edge rounded and corroded; and fragmented individuals are those that are broken with angular edges (Supplementary Fig. 3). In each sample, we documented the presence or absence of cave chalk. Where present, we categorized the surface condition of individual cave chalk fragments as either pristine or weathered. Pristine fragments are small, fragile and have a delicate floret structure; weathered fragments are larger and more rounded, with no floret structure.

**Data availability.** Data and modelling codes that have contributed from the reported results are available from the corresponding author at request.

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

## Acknowledgements

The research was supported by the National Research Foundation Singapore and the Singapore Ministry of Education under the Research Centers of Excellence initiative (grant M443B50147), a National Science Foundation award (EAR #0809392, 0809417, 0809625) and the Asia Research Institute and National University of Singapore. We are grateful to Simon Engelhart, Andrea Hawkes, Harvey Kelsey, Jedrzej (Yen) Majewski, Adam Switzer and Christopher Vane for helpful discussions and field support from Y. Ramayati, Aceh Heritage Community and T. Djubiantono, Pusat Penelitian dan Pengembangan Arkeologi Nasional. Lastly, we thank the people in the earthquake-affected region for their willingness to share their observations. This paper is a contribution to PALSEA2 (Paleo-Constraints on Sea-Level Rise 2) and to International Geoscience Programme (IGCP) Project 639, 'Sea Level Change from Minutes to Millennia'. This work comprises Earth Observatory of Singapore contribution no. 144. This research is supported by the National Research Foundation Singapore and the Singapore Ministry of Education under the Research Centres of Excellence initiative.

## Author contributions

C.M.R. oversaw all aspects of the research and led field work. C.M.R., B.P.H., K.S., J.P. and P.D. guided the intellectual direction of the research. C.M.R., B.P.H., K.S., P.D. and N.I. assisted with field work. N.I. and P.D. provided logistical support for the field work. C.M.R., B.P.H., K.S., J.P. and P.D completed the stratigraphical, sedimentological, foraminiferal and/or radiocarbon analyses. A.P. and B.P.H. conducted the Bayesian statistical analysis for the geochronology. Selected portions of the manuscript and/or supplement were written by all authors. All authors reviewed the manuscript.
