## [Peer Review File · Nature Communications]

Reviewers' Comments:

Reviewer #1 (Remarks to the Author)

Review of Manuscript

Highly variable recurrence of tsunamis in the 7500 years prior to the 2004 Indian Ocean Tsunami by Charles M. Rubin, Benjamin P. Horton, Kerry Sieh, Jessica E. Pilarczyk, Patrick Daly, Nazli Ismail, Andrew Parnell

The manuscript presents a well-researched and well-illustrated paleotsunami record from a spectacular site: a coastal cave in North Aceh, Indonesia, an area that was severely hit by the 2004 Indian Ocean tsunami. The main finding presented by the authors is a highly variable recurrence interval of past tsunamis between 2900 to 7400 years ago including a long dormant period of more than 2000 years.

These results are refining and extending a nearby paleoseismic record covering the time interval of 8000 to 3800 years ago by Kelsey et al. (2015), therefore furthering our understanding of tsunami hazards in the Northern Indian Ocean. Understanding the frequency and size of past tsunamis is essential for coastal communities in Aceh and elsewhere in the Indian Ocean region in order to prepare for future events and allocate resources accordingly.

In addition, a major contribution of this paper is the introduction of a new archive – caves – into paleotsunami work. To my knowledge this is the first time that a paleotsunami record has been discovered in a cave. The methods applied by the researchers in this study are state of the art and comprise new techniques, e.g. the use of cave chalk deposits. They are of great interest to the wider tsunami research community and could motivate further studies in similar geologic settings along active margins.

For the reasons outlined above I recommend this study for publication but have a few comments that could help clarify certain aspects of the manuscript.

General comments:

1. I am uncertain about the hydrodynamic situation at this coastal site and how sea level variations, high tide events and varying energetic conditions during the West and East Monsoon as well as coastal growth patterns might influence sedimentation in the cave. Currently the cave entrance lies only 1 m above Mean Tidal Level (MTL) and 120 m landward of the shoreline. I would think that sedimentation in the cave could be influenced by higher energetic events other than tsunamis. I think that a more-in-depth discussion of the hydrodynamic parameters and their variability is essential in understanding the sedimentary environment of the cave and the impact of tsunami events compared to background sedimentation.

2. The characteristics of the 11 sand layers are summarized in the main text and described in detail in the supplementary information which I found very helpful. The authors have convincing arguments that these indeed represent paleotsunami deposits. A key figure summarizing the characteristics of tsunami sands is extended figure 4, which should be moved into the main text. What remains a little unclear are the sedimentary processes that lead to the intercalated deposits between tsunami sands. The main text describes these deposits but offers little explanation for their formation. From the supplementary information I understand that some represent guano beds and deposits from ponding water in the cave after high precipitation events. Other intercalated beds are described as "marine-influenced clay" or fine sand that contains reworked foraminifera from underlying tsunami sands. Which processes rework tsunami sands and/or lead to

marine-influenced clay in the cave? This discussion has to be included in the main text to give the reader a better understanding of what happens in the cave during periods of quiescence. If possible, I suggest to expand extended figure 4 and include the characteristics of intercalated layers.

3. A main claim of this manuscript is a long dormant period between 5500 and 3400 years ago. A thin organic-rich layer between tsunami sands f and g represents this interval. I think the following questions should be discussed regarding this long dormant period:

a) Why is the intercalated layer between tsunami sands f and g so thin compared to other background sedimentation layers representing only 681 years?

b) The argument is made that the accommodation space in the cave is filled after deposition of deposit k at ~2900 years ago. Can a similar argument be made for the time interval between 5500 and 3400 years ago?

c) Is there a hiatus in the cave's sedimentary record due to lack of accommodation space or for other reasons, which would cause potential tsunami events to not be recorded during this time interval?

d) Kelsey et al. (2015) suggest a recurrence interval of 600-900 years for the time interval between 7400 and 3800 years ago. How does this fit your findings? There is a discrepancy between the dates of the upper buried soil in Kelsey et al. (2015) (~3800 years ago) and the date plotted in figure 4 (~3500 years ago). I am not sure how this might have influenced your age model.

4. The broader discussion at the end of the manuscript could be expanded. For example, I am curious how this paleotsunami record fits other mid-Holocene Indian Ocean tsunami records for example from Sri Lanka (Jackson et al. 2014, and data presented by Ranasinghe et al. at the 2015 GSA meeting). Also, is the observed pattern of a long dormant period followed by a cluster of smaller events been observed elsewhere? I would be cautious in suggesting that a long dormant period might occur after the 2004 event.

Minor comments:

Labeling of units 1-12 in the extended figures and supplementary text is a little confusing. I would avoid using Unit 1 etc. for the whole stratigraphic package and again for a subunit of Unit 1. Better use 1.0 etc. for subunits.

I am missing extended figures 1c,d,e and 2c.

Extended figure 4 should be in the main text. Could characteristics of layers between tsunami sands be included in this figure?

Thank you for the opportunity to review this exciting study!

Katrin Monecke, Wellesley College, Wellesley, MA, USA

Reviewer #2 (Remarks to the Author)

The manuscript "Highly variable recurrence of tsunamis in the 7,500 years prior to the 2004 Indian Ocean Tsunami" by Charles M. Rubin and colleagues presents a long and highly interesting sedimentary record of tsunami flooding in Aceh Province (NW Sumatra). The manuscript is concise and well written, and has important implications on our understanding of long-term (Holocene scales) seismic behaviour of one of the most dangerous plate boundaries worldwide. The authors provide evidence for much more irregular recurrence intervals of major tsunamigenic earthquakes along the Sunda Megathrust than previously assumed and ascribe this pattern to interfingering of large slip ruptures, earthquake super cycles and smaller slip failures. This paper will be of very high interest to researchers in (palaeo-)seismology, (palaeo-)tsunami research and regional coastal hazard management and probably will have significant impact on how researchers evaluate

palaeo-tsunami records for coastal hazard assessment in general.

However, I recommend that the authors address the following comments before publication:

P2L3-6: The authors use so far poor chronological constraints and the fragmented distribution of palaeotsunami records from Thailand and Sumatra to stress the importance of their record. I absolutely do not doubt its relevance and the impact it will have on understanding recurrence patterns of major tsunami along the Sunda Arc, in particular due to the wide time window it covers. However, I disagree with the statement: "...the timeline of their reconstructions is fragmentary given a lack of well-preserved tsunami deposits, preventing the long-term analysis of tsunami recurrence", in light of some very solid chronologies from spatially very well resolved deposits in Thailand based on OSL elaborated by Brill et al. (2012a,b,) and Prendergast et al. (2012), which, unfortunately were not cited by the authors. All studies cited to substantiate fragmentary timelines are based on a low to moderate number of indirect radiocarbon datings associated with a higher degree of uncertainty than OSL (changing reservoir effects over time, reworking etc.).

By the way, the exclusion of these papers underlines a certain regional disbalance in cited references. The manuscript contains a lot of self-citation, which basically is OK as the group of authors has worked a lot in the wider study area, but also thematic, non-regional references are almost exclusively from US researchers, while contributions of first authors from Asia, Europe or Australia are unequivocally underrepresented.

P4L13-16: Statement (3) is not entirely convincing, as the sources cited for a regional depth of the storm-wave base <20 m (Monecke et al., 2008; Sieh et al., 2015) provide no specific evidence: Sieh et al (2015) just state it (no evidence) and Monecke et al. (2008) do not explicitly refer to the storm-wave base. Please provide more substantial evidence for the regional storm wave base.

Presentation of data: In order to get published in a medium such as Nature Communications, data documentation in the Supplement material should be a bit more transparent and reader-friendly. Stratigraphic descriptions in the Supplement are very detailed, which is good, but I also suggest to visualize grain-size data by adding proxy columns of at least mean grain size to Extended Data Figure 4. At least fining-up patterns of the thick sand beds need to be supported by data, in particular to substantiate the important statement on page 4 "The normal grading suggests that each bed resulted from a single rather than multiple complete draining and filling of the cave with water". Furthermore, it should be demonstrated how the authors defined taphonomical categories, for instance by adding SEM pictures such as in Fig. 2 of Pilarczyk et al. (2016). Standard multivariate statistical analysis of foram data (e.g. PCA) would help the reader to grasp differences between sand beds, mud laminae and guano, or even the 3-wave-pulse-pattern on the 2004 at a glance.

Supplementary Material: More details are required for description of methods. How was grain-size data generated? What is the sampling resolution and what pre-treatment and analytical technique did the authors use? How much sample material was prepared for foram counting, how many specimens were counted per sample? etc.

Controls of deposit thickness: The influence of relative sea levels on the formation of tsunami deposits and inferences on slip patterns is in my opinion not sufficiently discussed. Even though I can follow the line of arguments in the Supplementary Material that "the presence of guano-rich beds in the coastal cave sequence suggests that RSL was below the lip of the cave entrance during their deposition", RSL (and sediment availability or lateral shoreline changes(?)), which is not discussed at all) might also play a role in the resulting thickness of the event layer – not just the size of slip ruptures. In my opinion, these factors at least need to be mentioned in the main text with reference to a more thorough discussion in the Supplementary Material

Minor comments:

P1: Which author is affiliated with University College Dublin?

P2L20-22: As word counts matter in this case, I suggest to condense this part. Very similar things have been stated at the end of the last section.

P3L2: Sunda Megathrust (consistently upper case)

P3L12: deposit – singular

P6L1: wording

P6L5: tsunami deposits

P6L7: framework

Ref1: 349–374 (copy/paste error...)

Ref2: Journal title abbreviation

Ref3: C.-C.

Ref4: ...Natawidjaja, Uplift...

Ref6: C. A. Grand Pre; B. P. Horton; Journal title abbreviation

Ref7: K. Sieh, P. Daly

Ref8: ...Prendergast, Medieval...; 1228–1231 (copy/paste error...)

Ref9: ...N. Cahill, Accomodation...

Ref10: (2015).

Ref12: Sed. Geol.

Ref13: Chagué-Goff; ...B. Jaffe, D. Dominey-Howes, Progress...

Ref15: wrong citation: chapter from the book Global Perspectives on Tropical Cyclones edited by Elsberry, published by WMO

Ref16: Journal title abbreviation

Ref17: C. Goldfinger; (2015)

Ref19: Z. Peng, J. Gomberg; journal title abbreviation

Ref21: I. Shennan, A. J. Long

Ref22: ...J. Bourgeois, Vented...; 59, 419-444

Ref23: 1248 (2001).

Ref24: Mar. Geol.

Ref25: J. C. Gomez, P. C. Rieg

Ref27: (2013)

Ref28: W. W.-S. Yim; 44, 167

Ref29: X.-L. Meng

Ref30: P. L. Gibbard;. Holocene 15

Overall rating based on the journal's criteria for publication:

"The data is technically sound" – Yes, for most parts. Overall stratigraphical descriptions and chronological modelling are well depicted, but there is definitely a need for more through data documentation in grain-size distribution, high-resolution bedding and foram analysis (see review below). Unfortunately, based on my limited background, I cannot comment on the Bayesian analysis of the 14C data.

The paper provides strong evidence for its conclusions – Yes, even though some factors, which may also control thickness of the tsunami deposits (relative sea level, sediment availability, shoreline changes over time) are not yet sufficiently discussed.

The results are novel (we do not consider abstracts and internet preprints to compromise novelty) – Yes.

The manuscript is important to scientists in the specific field – Yes.

In general, to be acceptable, a paper should represent an advance in understanding likely to

influence thinking in the field – Yes, I am sure it will.

Refs used in this review

Brill, D., et al., 2012a. *Quat. Geochronol.* 10, 224–229.

Brill, D., et al., 2012b. *Nat. Haz. Earth Syst. Sci.* 12, 2177–2192.

Pilarczyk, J.E., et al., 2016. *Mar. Geol.* 339, 104–114.

Prendergast, A., et al., 2012. *Mar. Geol.* 295–298, 20–27.

Author Responses to Reviews' Comments

Here we provide a list of the review comments, and the steps taken to address them in the revised draft of “*Highly variable recurrence of tsunamis in the 7,500 years prior to the 2004 Indian Ocean tsunami.*” Reviewer comments are identified in italics, followed by our indented responses.

REVIEWER 1 COMMENTS

Reviewer 1 Comment: I am uncertain about the hydrodynamic situation at this coastal site and how sea level variations, high tide events and varying energetic conditions during the West and East Monsoon as well as coastal growth patterns might influence sedimentation in the cave. Currently the cave entrance lies only 1 m above Mean Tidal Level (MTL) and 120 m landward of the shoreline. I would think that sedimentation in the cave could be influenced by higher energetic events other than tsunamis. I think that a more-in-depth discussion of the hydrodynamic parameters and their variability is essential in understanding the sedimentary environment of the cave and the impact of tsunami events compared to background sedimentation.

Author Response: This is an important point for establishing the context of the deposits in the cave. To further clarify we added several sentences [Lines 118 - 125] demonstrating that exposure to tropical cyclones is limited along the Aceh coast, eliminating the most likely alternative event for the sand deposits. This is supported by additional references. We have added additional interpretation of how the cave deposits were formed [Lines 80-9] and included a new figure [Figure 7].

We have added a subsection on ‘*Holocene sea levels and tsunami deposition in the coastal cave*’ [Lines 144-162] demonstrating that during the time span represented by the cave deposits (from mid Holocene until present) the cave entrance was always above the mean tidal level. We are fairly confident that the marine sand deposits are tsunami – and not the result of other coastal processes. We hope we have clarified this sufficiently in the main text, and in the detailed sediment descriptions which have been moved from the Supplementary Material to the Methods section.

Reviewer 1 Comment: The characteristics of the 11 sand layers are summarized in the main text and described in detail in the supplementary information which I found very helpful. The authors have convincing arguments that these indeed represent paleotsunami deposits. A key figure summarizing the characteristics of tsunami sands is extended figure 4, which should be moved into the main text.

Author Response: Given the expanded space offered by the Nature Communications format, we have moved most of the detailed description of the sediment beds from the supplementary section to the main text. Extended Figure 4 has been moved to the main text as Figure 5. We have modified Figure 5 to highlight grain size and modified Supplementary Tables 1 & 2 that highlight grain size.

Reviewer 1 Comment: What remains a little unclear are the sedimentary processes that lead to the intercalated deposits between tsunami sands. The main text describes these deposits but offers little explanation for their formation. From the supplementary information I understand that some represent guano beds and deposits from ponding water in the cave after high precipitation events. Other intercalated beds are described as “marine-influenced clay” or fine sand that contains reworked foraminifera from underlying tsunami sands. Which processes rework tsunami sands and/or lead to marine-influenced clay in the cave? This discussion has to be included in the main text to give the reader a better understanding of what happens in the cave during periods of quiescence. If possible, I suggest to expand extended figure 4 and include the characteristics of intercalated layers.

Author Response: We have added a sub-section ‘*Intercalated beds*’ [Lines 80 - 91] to provide more details about the formation processes of the intercalated beds. Interpretation of the clay deposits is discussed in Lines 92-98, as well as in the Methods.

We briefly discuss sedimentation process in the cave between tsunamis [Lines 115-119]. In Figure 5 we highlight the foram abundances and taphonomy, and the characteristics of the cave chalk. We also added an SEM image of the forams [Supplementary Figure 3]. Detailed assessment of grain size for the intercalated beds is included within Supplementary Tables 1 & 2.

Reviewer 1 Comment: A main claim of this manuscript is a long dormant period between 5500 and 3400 years ago. A thin organic-rich layer between tsunami sands f and g represents this interval. I think the following questions should be discussed regarding this long dormant period: a) Why is the intercalated layer between tsunami sands f and g so thin compared to other background sedimentation layers representing only 681 years?

Author Response: The thickness of the guano layer between F and G is better visualized in Supplementary Figures 1 & 2, where the deposit is thicker. We don’t expect much accumulation in the cave given the limited depositional processes, and the breakdown of the guano addressed in Lines 83-87.

Reviewer 1 Comment: The argument is made that the accommodation space in the cave is filled after deposition of deposit k at ~2900 years ago. Can a similar argument be made for the time interval between 5500 and 3400 years ago?

Author Response: This is an important point. We added Lines 112-118 & 154-162 and an illustrative Figure 7 to show that the 2004 event scoured out sediment which was situated above the level of a rock sill in the cave. There is clear evidence for this erosion at the interface between the 2004 sand and the underlying deposit.

We suggest that deposits under the 2,900PB deposit were able to accumulate as they were protected from similar erosion by the shape of the cave and rock fall. Additionally, there is no evidence for significant scouring between sand layer F and G (which can be seen in Supplementary Figures 1 & 2, as well as in the detailed description of the units in the

Methods). We emphasize that utilizing coastal caves is a new addition to tsunami science, and offers great potential for finding well preserved tsunami deposits.

Reviewer 1 Comment: Is there a hiatus in the cave's sedimentary record due to lack of accommodation space or for other reasons, which would cause potential tsunami events to not be recorded during this time interval?

Author Response: This is addressed in part in the above response. Additionally, in Lines 214-216 we acknowledge that it is possible that factors such as sediment source availability or lateral shoreline changes may lead to some tsunami not being recorded in the cave. However, given the well preserved and continuous stratigraphy (Supplementary Figures 1 & 2), we think it is reasonable that tsunami over a certain size would leave some sediment record.

Reviewer 1 Comment: Kelsey et al. (2015) suggest a recurrence interval of 600-900 years for the time interval between 7400 and 3800 years ago. How does this fit your findings? There is a discrepancy between the dates of the upper buried soil in Kelsey et al. (2015) (~3800 years ago) and the date plotted in figure 4 (~3500 years ago). I am not sure how this might have influenced your age model.

Author Response: We have included a new subsection ‘*Variable recurrence of tsunami*’ discussing the large variations of recurrence intervals along subduction zones.

The reviewer is correct that in our Table 1 the upper buried soil has a calibrated age of 3,800 years BP (3717 – 3902). However, we derive the age of the Sand bed G from the charcoal recovered from the cave ((3362-3246 years BP). The charcoal age is supported by the buried soil, which provide a maximum limiting age. We have clarified this confusion with an improved Table 1 caption.

Reviewer 1 Comment: The broader discussion at the end of the manuscript could be expanded. For example, I am curious how this paleotsunami record fits other mid-Holocene Indian Ocean tsunami records for example from Sri Lanka (Jackson et al. 2014, and data presented by Ranasinghe et al. at the 2015 GSA meeting).

Author Response: We have added material and references for the mid-Holocene Indian Ocean tsunami records for India, Sri Lanka, the Maldives and Thailand in the introduction section to provide a wider geographical context for our work Lines 24 -30.

We have added a sub-section in the discussion ‘*Comparison with other Indian Ocean prehistoric tsunami records*’ that provide context to the wider regional record provided from these other sites Lines 176-189. While we do not belabour it, we are not fully confident that the data from the Maldives and Sri Lanka clearly represent tsunami deposits from our review of the published work. We approach this issue cautiously, and propose that the proximity of the Aceh cave site to the Sunda Megathrust makes it a more reliable indicator of tsunamis generated by ruptures of this fault that sites located in South

Asia, where there is the possibility for deposits to result from storm, and/or tsunami generated by other faults [Lines 187-189].

Reviewer 1 Comment: Also, is the observed pattern of a long dormant period followed by a cluster of smaller events been observed elsewhere? I would be cautious in suggesting that a long dormant period might occur after the 2004 event.

Author Response: This long dormant period has not been recorded before but many studies record variable recurrence intervals. We have added text to discuss one such paper that shows a similar large variation of recurrence intervals from 56 years to about 550 years (Meltzner et al., 2010) [Lines 205-209]. We specifically state that we ‘suggest’ the possibility of a dormant period, and avoid using words such as ‘demonstrate’. We end the paper with a note of caution that hazard mitigation plans should focus on the inevitability of future tsunami, rather than trying to anticipate possible intervals between them [Lines 232-235].

Reviewer 1 Comment: Labeling of units 1-12 in the extended figures and supplementary text is a little confusing. I would avoid using Unit 1 etc. for the whole stratigraphic package and again for a subunit of Unit 1. Better use 1.0 etc. for subunits.

Author Response: We have corrected errors in the supplement, which caused confusion. We use units and sub-units. Tsunami layers are layered A – K.

Reviewer 1 Comment: I am missing extended figures 1c,d,e and 2c.

Author Response: These numbers represent boxes within the main body of the figures, and not additional figures. We have made this more explicit within the caption of the figures.

Reviewer 1 Comment: Extended figure 4 should be in the main text. Could characteristics of layers between tsunami sands be included in this figure?

Author Response: Figure 4 has been moved to the main text as requested. The figure has been expanded to include characteristics of the tsunami beds.

REVIEWER 2 COMMENTS

Reviewer 2 Comment: P2L3-6: The authors use so far poor chronological constraints and the fragmented distribution of palaeotsunami records from Thailand and Sumatra to stress the importance of their record. I absolutely do not doubt its relevance and the impact it will have on understanding recurrence patterns of major tsunami along the Sunda Arc, in particular due to the wide time window it covers. However, I disagree with the statement: "...the timeline of their reconstructions is fragmentary given a lack of well-preserved tsunami deposits, preventing the long-term analysis of tsunami recurrence", in light of some very solid chronologies from spatially very well resolved deposits in Thailand based on OSL elaborated by Brill et al. (2012a,b,) and Prendergast et al. (2012), which, unfortunately were not cited by the authors. All studies cited to substantiate fragmentary timelines are based on a low to moderate number of indirect radiocarbon datings associated with a higher degree of uncertainty than OSL (changing reservoir effects over time, reworking etc.).

By the way, the exclusion of these papers underlines a certain regional disbalance in cited references. The manuscript contains a lot of self-citation, which basically is OK as the group of authors has worked a lot in the wider study area, but also thematic, non-regional references are almost exclusively from US researchers, while contributions of first authors from Asia, Europe or Australia are unequivocally underrepresented.

Author Response: Very important points – we have amended our discussion of paleotsunami sites in the introduction of the paper to include sites from India, Sri Lanka and the Maldives to address our regional bias [Lines 24-30]. We have added explicit comparative discussion of these regional sites in a new subsection ‘*Comparison with other Indian Ocean prehistoric tsunami records*’ in the Discussion [Lines 176-189].

This better contextualizes how the cave deposits are situated within wider regional records. We used the increased size offered by Nature Communications to double our reference list, which allows us to broaden our discussion about wider regional studies.

We have tried throughout the paper to better demonstrate how the data from the Aceh cave contributes towards regional records.

Reviewer 2 Comment: P4L13-16: Statement (3) is not entirely convincing, as the sources cited for a regional depth of the storm-wave base <20 m (Monecke et al., 2008; Sieh et al., 2015) provide no specific evidence: Sieh et al (2015) just state it (no evidence) and Monecke et al. (2008) do not explicitly refer to the storm-wave base. Please provide more substantial evidence for the regional storm wave base.

Author Response: The reviewer’s point is well taken. We removed Monecke and Sieh references and added Anthes 1982 and McBride 1995 to discuss cyclone tracks in the region. We consulted with a number of tsunami modellers, and found that they generally use a 20m wave-base as an estimate, but lack a reliable empirical basis for this off the coast of western Sumatra. Therefore, we have removed the text concerning wave base and replaced it with a discussion of the foraminiferal habitat, which is more relevant to

our analysis.

Reviewer 2 Comment: Presentation of data: In order to get published in a medium such as Nature Communications, data documentation in the Supplement material should be a bit more transparent and reader-friendly. Stratigraphic descriptions in the Supplement are very detailed, which is good, but I also suggest to visualize grain-size data by adding proxy columns of at least mean grain size to Extended Data Figure 4.

Author Response: We have used the additional space offered by the Nature Communication format to include the stratigraphic descriptions into the Methods section of the main text. This will ensure that all relevant material and data is included in the PDF download of the paper. We have moved Extended Figure 4 into the main text as Figure 5.

We adjusted Figure 5 to include mean grain sizes that contain either coarse pulses or fining upwards sequences as both reviewers have requested. Detailed data on grain size for each unit is included within supplementary tables 1 & 2.

Reviewer 2 Comment: At least fining-up patterns of the thick sand beds need to be supported by data, in particular to substantiate the important statement on page 4 “The normal grading suggests that each bed resulted from a single rather than multiple complete draining and filling of the cave with water”.

Author Response: The fining upwards patterns are supported by data provided in Supplementary Tables 1 & 2. In addition, Fig. 3 (2004 tsunami deposits) has been moved into the main text. We have moved all of the sediment descriptions, which contains detailed information about the sorting of the sand beds, from the Supplementary Information into the Methods section – which will make this material much more accessible to the reader.

Reviewer 2 Comment: Furthermore, it should be demonstrated how the authors defined taphonomical categories, for instance by adding SEM pictures such as in Fig. 2 of Pilarczyk et al. (2016).

Author Response: The three taphonomic categories of foraminifera are now described in the supplementary methods. In addition, the new Supplementary Figure 3 presents SEM images of each taphonomic category.

Reviewer 2 Comment: Standard multivariate statistical analysis of foram data (e.g. PCA) would help the reader to grasp differences between sand beds, mud laminae and guano, or even the 3-wave-pulse-pattern on the 2004 at a glance.

Author Response: It is presence vs. absence of foraminifera that distinguishes sand beds from intercalating units, not ecology. Because intercalating units sometimes contain

small concentrations of reworked foraminifera, multivariate statistical analyses to not produce clear separation. Instead, we graphically highlight the sharp differences in presence vs. absence in Figure 4 in the main text.

Reviewer 2 Comment: Supplementary Material: More details are required for description of methods. How was grain-size data generated? What is the sampling resolution and what pre-treatment and analytical technique did the authors use?

Author Response: A subsection ‘Grain size analysis’ has been included in the Methods providing more detail about how the grain size was analysed [Lines 602-610].

Reviewer 2 Comment: How much sample material was prepared for foram counting, how many specimens were counted per sample? etc.

Author Response: A subsection ‘Foraminiferal and cave chalk analysis’ has been included in the Methods providing more detail about how the forams were analysed [Lines 612-627].

Reviewer 2 Comment: Controls of deposit thickness: The influence of relative sea levels on the formation of tsunami deposits and inferences on slip patterns is in my opinion not sufficiently discussed. Even though I can follow the line of arguments in the Supplementary Material that “the presence of guano-rich beds in the coastal cave sequence suggests that RSL was below the lip of the cave entrance during their deposition”, RSL (and sediment availability or lateral shoreline changes(?), which is not discussed at all) might also play a role in the resulting thickness of the event layer – not just the size of slip ruptures. In my opinion, these factors at least need to be mentioned in the main text with reference to a more thorough discussion in the Supplementary Material.

Author Response: These are two important points. We have added a sub-section ‘Holocene sea levels tsunami deposition in the coastal cave’ to the main text to support our argument that relative sea level was most likely below the entrance level of the cave for the depositional period represented in the cave record [Lines 144-162].

In the main text we acknowledge that sediment availability and lateral shoreline changes might also contribute to the thickness of the layers, and argue for what we feel is the most likely facts determining depositional characteristics [Lines 212-216].

Reviewer 2 Comment: P1: Which author is affiliated with University College Dublin?

Author Response: Corrected in text – Andrew Parnell.

Reviewer 2 Comment: P2L20-22: As word counts matter in this case, I suggest to condense this part. Very similar things have been stated at the end of the last section.

Author Response: The text is within the word count suitable for Nature Communication. However, we have tried to limited redundancies in the main text.

Reviewer 2 Comment: P3L2: Sunda Megathrust (consistently upper case)

Author Response: Addressed throughout the text and figures.

All minor editorial comments made by Reviewer 2 have been addressed in the revised text.

Reviewers' Comments:

Reviewer #1:

Remarks to the Author:

I find that the authors addressed questions and concerns thoroughly and feel that the sedimentary record of the cave has become much clearer. I specifically like that Figure 5 is now part of the main text since it beautifully summarizes the main findings and supports the interpretation of the suggested tsunami deposits. With the newly added section, the depositional process of intercalated layers is much clearer. The discussion has gained more weight by comparing this record to other archives in the Indian Ocean region and by a more-in-depth discussion of the rupture behavior. It is wise to caution against any diminishing tsunami preparedness efforts for this region. Overall, this manuscript does indeed greatly enhance our understanding of the rupture behavior along this megathrust and provides an exciting new archive for paleotsunami research.

Reviewer #2:

Remarks to the Author:

Issues raised in the first round of reviews have been well-addressed by the authors. In particular, I appreciate that the manuscript was extended and that the research was put in a wider regional context. I am certain that the conclusions drawn from the analysis of this fascinating record and that the paper will have a significant influence on paradigms in palaeotsunami research and the long-term seismic performance of megathrusts.

Minor comments:

L142: Something seems to be wrong with the predicate here(?)...

Fig. 5: In the legend, "channeling" has two different abbreviations.

Data presentation: I still think that a down-profile depiction of grain size data (mean, sand content) in Fig. 5 would be more reader-friendly than to just show arrows for normal grading. I am aware that the supplement file has long, detailed tables of grain size data, but this is hard to read and not in context with other stratigraphic characteristics as in Fig. 5.

Max Engel

University of Cologne, Germany

REVIEWER 1 COMMENTS

Reviewer #1 (Remarks to the Author):

I find that the authors addressed questions and concerns thoroughly and feel that the sedimentary record of the cave has become much clearer. I specifically like that Figure 5 is now part of the main text since it beautifully summarizes the main findings and supports the interpretation of the suggested tsunami deposits. With the newly added section, the depositional process of intercalated layers is much clearer. The discussion has gained more weight by comparing this record to other archives in the Indian Ocean region and by a more-in-depth discussion of the rupture behavior. It is wise to caution against any diminishing tsunami preparedness efforts for this region. Overall, this manuscript does indeed greatly enhance our understanding of the rupture behavior along this megathrust and provides an exciting new archive for paleotsunami research.

No revisions are necessary

REVIEWER 2 COMMENTS

Reviewer #2 (Remarks to the Author):

Issues raised in the first round of reviews have been well-addressed by the authors. In particular, I appreciate that the manuscript was extended and that the research was put in a wider regional context. I am certain that the conclusions drawn from the analysis of this fascinating record and that the paper will have a significant influence on paradigms in palaeotsunami research and the long-term seismic performance of megathrusts.

No revisions are necessary

Minor comments:

L142: Something seems to be wrong with the predicate here(?)...

Fig. 5: In the legend, "channeling" has two different abbreviations.

Data presentation: I still think that a down-profile depiction of grain size data (mean, sand content) in Fig. 5 would be more reader-friendly than to just show arrows for normal grading. I am aware that the supplement file has long, detailed tables of grain size data, but this is hard to read and not in context with other stratigraphic characteristics as in Fig. 5.

Author Response:

L142: We clarified and slightly modified the offending syntax on L142.

Fig. 5 We corrected the abbreviations in Fig 5., so the abbreviations are consistent.

Fig. 5 data presentation . We adjusted Figure 5 to include mean grain sizes that contain either coarse pulses or fining upwards sequences as arrows. Given the size of Figure 5, we do not think that adding a color down profile depiction of the grain size data will be legible or useful to the reader. If the reader is more interested in the grain size data, it is including within supplementary tables 1 & 2.